# The long noncoding RNA lncCIRBIL disrupts the nuclear translocation of Bclaf1 alleviating cardiac ischemia–reperfusion injury

Yang Zhang[1,5], Xiaofang Zhang [1,5], Benzhi Cai [1,5], Ying Li[1], Yuan Jiang [1], Xiaoyu Fu [1], Yue Zhao [1], Haiyu Gao[1], Ying Yang[1], Jiming Yang[1], Shangxuan Li[1], Hao Wu [1], Xuexin Jin[1], Genlong Xue[1], Jiqin Yang[1], Wenbo Ma[1], Qilong Han[1], Tao Tian[1], Yue Li[2,3], Baofeng Yang [1,4✉], Yanjie Lu[1] & Zhenwei Pan [1,4✉]

Cardiac ischemia–reperfusion (I/R) injury is a pathological process resulting in cardiomyocyte death. The present study aims to evaluate the role of the long noncoding RNA Cardiac Injury-Related Bclaf1-Inhibiting LncRNA (lncCIRBIL) on cardiac I/R injury and delineate its mechanism of action. The level of lncCIRBIL is reduced in I/R hearts. Cardiomyocyte-specific transgenic overexpression of lncCIRBIL reduces infarct area following I/R injury. Knockout of lncCIRBIL in mice exacerbates cardiac I/R injury. Qualitatively, the same results are observed in vitro. LncCIRBIL directly binds to BCL2-associated transcription factor 1 (Bclaf1), to inhibit its nuclear translocation. Cardiomyocyte-specific transgenic overexpression of Bclaf1 worsens, while partial knockout of Bclaf1 mitigates cardiac I/R injury. Meanwhile, partial knockout of Bclaf1 abrogates the detrimental effects of lncCIRBIL knockout on cardiac I/R injury. Collectively, the protective effect of lncCIRBIL on I/R injury is accomplished by inhibiting the nuclear translocation of Bclaf1. LncCIRBIL and Bclaf1 are potential therapeutic targets for ischemic cardiac disease.

[1] Department of Pharmacology (The Key Laboratory of Cardiovascular Research, Ministry of Education) at College of Pharmacy, Harbin Medical University, Harbin 150086, China. [2] Institute of Metabolic Disease, Heilongjiang Academy of Medical Science, Harbin Medical University Cancer Hospital, Harbin, China. [3] Department of Cardiology, The First Affiliated Hospital, Harbin Medical University, Youzheng Street 23#, Nangang District, Harbin 150001 Heilongjiang Province, China. [4] Research Unit of Noninfectious Chronic Diseases in Frigid Zone, Chinese Academy of Medical Sciences, 2019RU070, Harbin, China. [5]These authors contributed equally: Yang Zhang, Xiaofang Zhang, Benzhi Cai. ✉email: yangbf@ems.hrbmu.edu.cn; panzw@ems.hrbmu.edu.cn

Cardiac ischemia–reperfusion (I/R) injury is a major contributor to the morbidity and mortality of patients with severe coronary artery disease. Despite of our continuous advancement in the understanding of the molecular mechanisms of this pathological process, no specific target-oriented therapy has been validated thus far, and the treatment of reperfusion injury following ischemia is still supportive[1–3]. There remains a great need to further explore the mechanisms of cardiac I/R injury and identify potential therapeutic targets.

Long noncoding RNAs (lncRNAs) are a large and diverse class of RNAs that are >200 nucleotides in length and possesses no protein-coding property[4]. They regulate diverse biological processes by altering gene expression networks in the nucleus and by modulating mRNA stability, translation, and protein function in the cytoplasm[5]. LncRNAs exhibit their regulatory actions on biological and pathological processes by directly interacting with RNA, DNA, or protein to modulate chromatin structures or serving as scaffolds or decoys for microRNAs[6–8]. Emerging researches suggest that lncRNAs are key regulators of various cardiac diseases, including cardiac I/R injury. A host of lncRNAs have been identified as important determinants of cardiac I/R injury, mostly by sponging cardiac injury-related miRNAs[6,7]. Considering the huge number of lncRNAs (96,308 lncRNA genes estimated in human genome by NONCODE) and their versatile modes of action in the cell, our current understanding of the regulation of cardiac I/R injury by lncRNAs is still rather limited. LncRNAs that have potential regulatory mechanisms in cardiac I/R injury remain to be discovered.

Bcl-2-associated transcription factor-1 (Bclaf1) is an apoptosis-regulating protein that was originally identified via screening proteins that can interact with the adenoviral bcl-2 homolog E1B19K. The Bcl-2 protein family members bind to Bclaf1 and prevent its localization to the nucleus[9]. Bclaf1 has been demonstrated to induce apoptosis by activating the p53 pathway. Overexpression of Bclaf1 induces apoptosis, which can be reversed by Bcl-2[9,10]. Bclaf1 can activate TP53 gene transcription by binding to the core promoter element in the promoter region[11]. Moreover, in cultured colon adenocarcinoma cells, Bclaf1 overexpression increases the levels of pro-apoptotic proteins, p53 and Bax[12–14]. Additionally, Bclaf1 plays critical roles in lung development[15], T cell activation[16], B cell development[17], and virus infection[18]. However, the role of Bclaf1 in cardiac diseases remains unexplored.

In order to identify lncRNAs involved in cardiac I/R injury, we screened the lncRNAs for their differential expression levels in the heart of mouse cardiac I/R model and discovered that lncRNA NONMMUT058343 was significantly downregulated. We therefore focused on the role of this lncRNA in cardiac I/R injury and the underlying molecular mechanism. Our data showed that lncRNA NONMMUT058343 protects the heart from I/R injury by binding to Bclaf1 to impede its nuclear translocation. For convenience, we named lncRNA UT058343 as cardiac injury-related bclaf1-inhibiting lncRNA (lncCIRBIL). Our findings imply that lncCIRBIL and Bclaf1 are potential therapeutic targets for ischemic cardiac disease.

## Results

**Transgenic overexpression of lncCIRBIL alleviates cardiac I/R injury in mice.** To screen for the lncRNAs involved in myocardial I/R injury, we performed lncRNA microarray analysis on cardiac tissues dissected from ischemia zone, border zone (BZ), and the remote non-ischemia zone (NIZ) of cardiac I/R mice. The analysis identified lncCIRBIL as one of the deregulated lncRNAs that were downregulated in both ischemic and BZ relative to remote NIZ (Fig. 1A). The level of lncCIRBIL was lower in the heart of I/R mice than sham controls (Fig. 1B). The data from NONCODE (http://www.noncode.org/) and UCSC (http://genome.ucsc.edu/) database revealed that lncCIRBIL is an intergenic lncRNA of 862 nucleotides (nts) long and it locates on chromosome 6 (Supplementary Fig. 1). By blasting this sequence with human genome, we found a conservative sequence motif (407 nts) of human lncCIRBIL (Supplementary Fig. 2A–C). The plasma level of the human homolog of lncCIRBIL was also lower in patients with acute myocardial infarction (AMI) than in non-AMI control subjects (Fig. 1C). To explore the role of lncCIRBIL in cardiac I/R injury, we generated cardiac-specific lncCIRBIL transgenic overexpression (lncCIRBIL Tg) mice (Supplementary Fig. 3) and established a mouse model of cardiac I/R injury. The expression of lncCIRBIL was significantly increased in lncCIRBIL Tg mice than in wild-type controls (Fig. 1D). LncCIRBIL overexpression significantly reduced infarct size in the heart of I/R mice (Fig. 1E) and improved cardiac function as indicated by the increased ejection fraction (EF) and fractional shortening (FS) (Fig. 1F). Lactate dehydrogenase (LDH) and creatine kinase isoenzyme-MB (CKMB) are enzymes residing in the cytoplasm of cardiomyocytes and are released into the plasma upon cardiomyocyte injury. The levels of plasma LDH and CKMB were elevated after I/R injury, and these changes were inhibited by overexpression of lncCIRBIL (Fig. 1G, H). In addition, overexpression of lncCIRBIL decreased caspase-3 activity, cleaved caspase-3 protein level, and TUNEL (TdT-mediated dUTP nick end labeling)-positive cells of cardiac I/R mice (Fig. 1I–K).

We further explored the regulation of lncCIRBIL on hypoxia–reoxygenation (H/R)-induced injury of cultured mouse cardiomyocytes. LncCIRBIL was significantly decreased in cardiomyocytes exposed to H/R, hypoxia, hydrogen peroxide ($H_2O_2$), or hypoxia with energy deprivation (low glucose) (Supplementary Fig. 4A–D). Transfection of the lncCIRBIL-overexpressing plasmid into cardiomyocytes increased the level of lncCIRBIL (Supplementary Fig. 4E) and remarkably reduced H/R-induced apoptosis as reflected by significant decreases in caspase-3 activity, cleaved caspase-3 protein level, and TUNEL-positive cells (Supplementary Fig. 4F–H). These data indicate that lncCIRBIL is a cardioprotective lncRNA against cardiac I/R injury both in vivo and in vitro.

**Deficiency of lncCIRBIL exacerbates cardiac I/R injury.** To investigate whether downregulation of lncCIRBIL is detrimental to the heart, we assessed the effects of lncCIRBIL deficiency on cardiac injury. We first established lncCIRBIL knockout mice by CRISPR/Cas9 (clustered regulatory interspaced short palindromic repeats/CRISPR-associated protein 9) techniques (Supplementary Fig. 5) and then created cardiac I/R injury. The level of lncCIRBIL was undetectable in lncCIRBIL knockout mice (Fig. 2A). LncCIRBIL knockout increased the infarct size and impaired cardiac function of I/R hearts (Fig. 2B, C). The plasma levels of LDH and CKMB were increased in mice with cardiac I/R injury and these deteriorative alterations were further exacerbated with knockout of lncCIRBIL (Fig. 2D, E). Consistently, the caspase-3 activity, cleaved caspase-3 protein level, and TUNEL-positive cells were all increased in the hearts of I/R mice and these pro-apoptotic changes were further augmented by lncCIRBIL knockout (Fig. 2F–H).

We then tested the role of lncCIRBIL knockdown in H/R-induced cardiomyocyte injury in vitro. Transfection of lncCIRBIL small interfering RNA (siRNA) significantly reduced the level of lncCIRBIL in cultured cardiomyocytes (Supplementary Fig. 6A). Caspase-3 activity, cleaved caspase-3 protein level, and TUNEL-positive cells were increased in H/R cardiomyocytes, which was aggravated by silencing lncCIRBIL expression (Supplementary

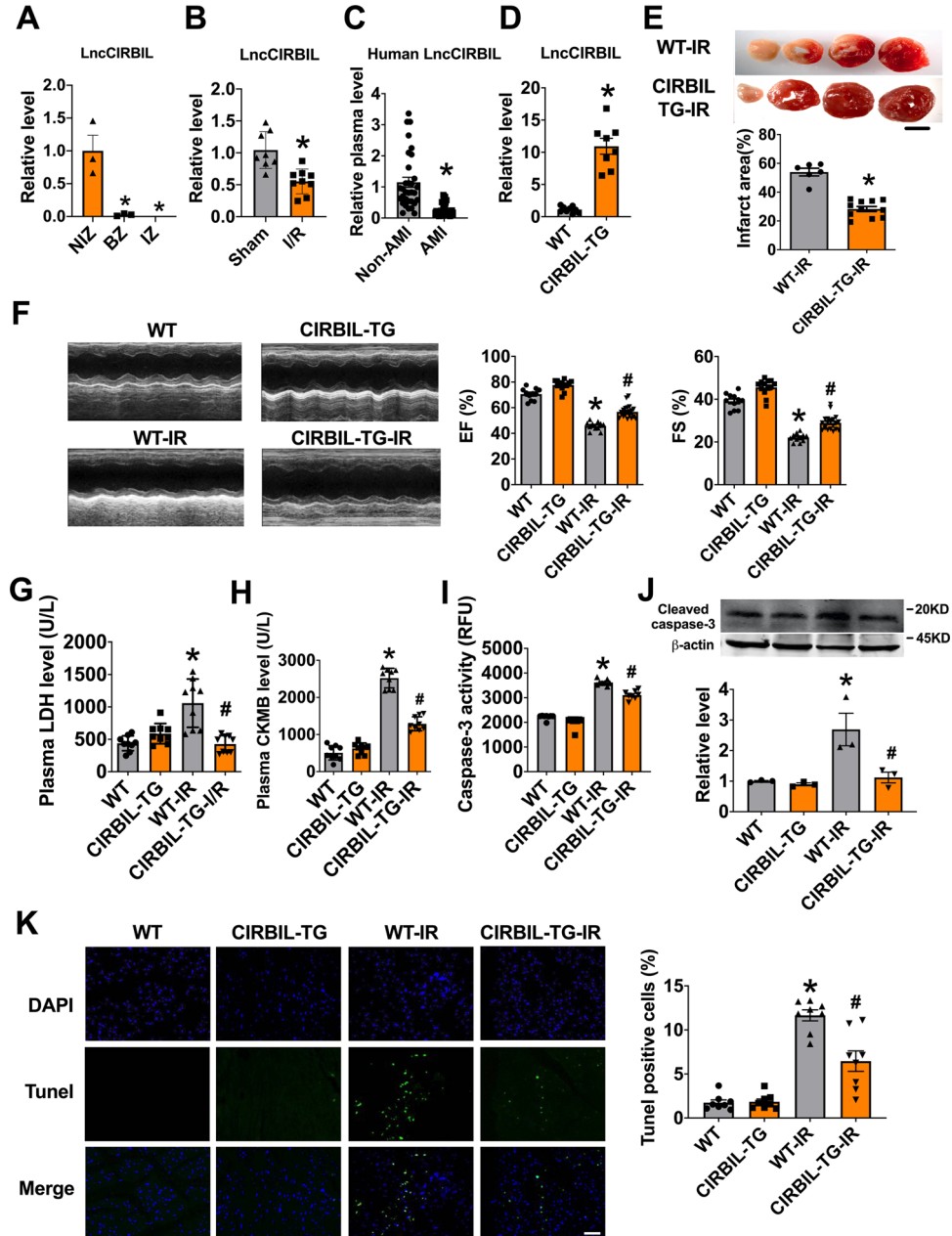

**Fig. 1 Cardiomyocyte-specific transgenic overexpression of lncCIRBIL alleviates cardiac ischemia–reperfusion injury. A** The levels of lncCIRBIL in ischemia zone (IZ), border zone (BZ), and non-ischemic zone (NIZ) of ischemia–reperfusion hearts. $N = 3$. *$P < 0.05$ versus NIZ. $P$ values were determined by one-way ANOVA test followed by Bonferroni post hoc analysis. **B** The level of lncCIRBIL in ischemia–reperfusion (I/R) and sham-operated hearts. Sham $N = 8$, I/R $N = 9$. *$P < 0.05$ versus Sham. $P$ values were determined by unpaired $t$ test. **C** Level of human lncCIRBIL in the plasma of acute myocardial infarction (AMI) patients. $N = 32$. *$P < 0.05$ versus non-AMI group. $P$ values were determined by unpaired $t$ test. **D** Level of lncCIRBIL in the heart of lncCIRBIL transgenic mice. WT $N = 9$, CIRBIL-TG $N = 8$. *$P < 0.05$ versus WT group. $P$ values were determined by unpaired $t$ test. **E** Infarct size of I/R hearts by TTC staining (scale bar: 2 mm). WT-I/R $N = 6$, CIRBIL-TG-I/R $N = 11$. *$P < 0.05$ versus WT-IR mice. $P$ values were determined by unpaired $t$ test. **F** Representative images of echocardiographs and statistics of ejection fraction (EF) and fractional shortening (FS). WT $N = 11$, CIRBIL-TG $N = 13$, WT-I/R $N = 11$, CIRBIL-TG-I/R $N = 21$. *$P < 0.05$ versus WT, #$P < 0.05$ versus WT-IR. $P$ values were determined by one-way ANOVA test followed by Bonferroni post hoc analysis. **G**, **H** Plasma levels of LDH and CKMB, respectively. $N = 9$ mice per group. *$P < 0.05$ versus WT, #$P < 0.05$ versus WT-IR. $P$ values were determined by one-way ANOVA test followed by Bonferroni post hoc analysis. **I** Caspase-3 activity. $N = 8$ mice per group. *$P < 0.05$ versus WT, #$P < 0.05$ versus WT-IR. $P$ values were determined by one-way ANOVA test followed by Bonferroni post hoc analysis. **J** Cleaved caspase-3 protein level. $N = 3$ mice per group. *$P < 0.05$ versus WT, #$P < 0.05$ versus WT-IR. $P$ values were determined by one-way ANOVA test followed by Bonferroni post hoc analysis. **K** Apoptosis of cardiomyocyte in the border zone (BZ) determined by TUNEL staining (scale bar: 20 μm). $N = 8$ mice per group. *$P < 0.05$ versus WT, #$P < 0.05$ versus WT-IR. $P$ values were determined by one-way ANOVA test followed by Bonferroni post hoc analysis. Data are expressed as mean ± SEM. WT wild type, WT-IR wild type with ischemia–reperfusion, CIRBIL-TG CIRBIL transgenic overexpression, CIRBIL-TG-IR CIRBIL transgenic overexpression mice with ischemia–reperfusion.

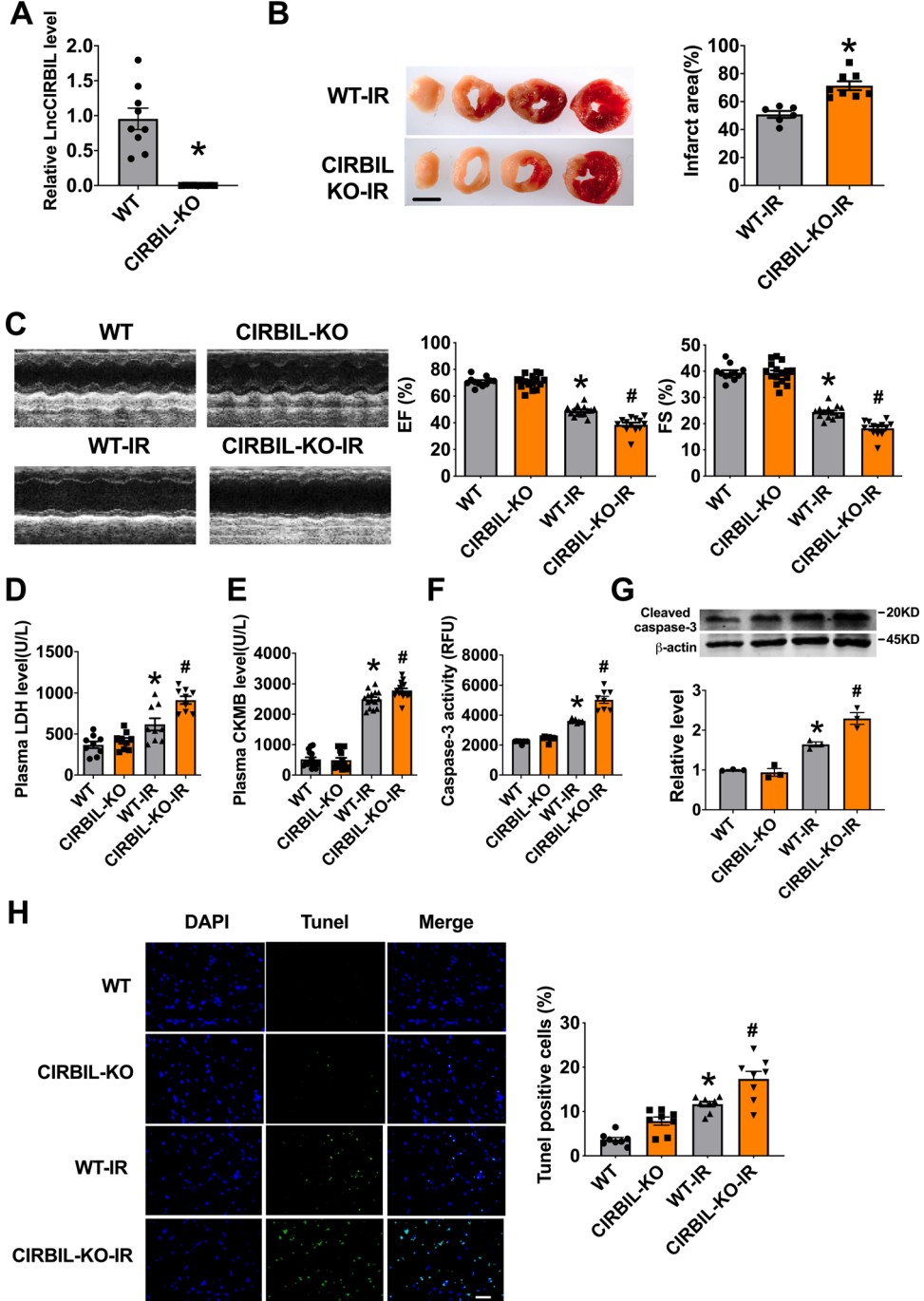

**Fig. 2 Knockout of lncCIRBIL exacerbates cardiac ischemia–reperfusion injury. A** LncCIRBIL levels in the hearts of lncCIRBIL knockout and wild-type (WT) mice. $N = 9$. *$P < 0.05$ versus WT group. $P$ values were determined by unpaired $t$ test. **B** Infarct size of I/R hearts stained with TTC (scale bar: 2 mm). WT-I/R $N = 6$, CIRBIL-KO-I/R $N = 8$. *$P < 0.05$ versus WT-IR mice. $P$ values were determined by unpaired $t$ test. **C** Representative images of echocardiographs and statistics of ejection fraction (EF) and fractional shortening (FS). WT $N = 11$, CIRBIL-KO $N = 16$, WT-I/R $N = 14$, CIRBIL-KO-I/R $N = 13$. *$P < 0.05$ versus WT; #$P < 0.05$ versus WT-IR mice. $P$ values were determined by one-way ANOVA test followed by Bonferroni post hoc analysis. **D** Serum concentrations of LDH, respectively. $N = 9$ mice per group. *$P < 0.05$ versus WT; #$P < 0.05$ versus WT-IR mice. $P$ values were determined by one-way ANOVA test followed by Bonferroni post hoc analysis. **E** Serum concentrations of CKMB, respectively. $N = 15$ mice per group. *$P < 0.05$ versus WT; #$P < 0.05$ versus WT-IR mice. $P$ values were determined by one-way ANOVA test followed by Bonferroni post hoc analysis. **F** Caspase-3 activity. $N = 8$ mice per group. *$P < 0.05$ versus WT-IR mice. $P$ values were determined by one-way ANOVA test followed by Bonferroni post hoc analysis. **G** Cleaved caspase-3 protein level. $N = 3$ mice per group. *$P < 0.05$ versus WT-IR mice. $P$ values were determined by one-way ANOVA test followed by Bonferroni post hoc analysis. **H** Apoptosis of cardiomyocyte in the border zone (BZ) by TUNEL staining (scale bar: 20 μm). $N = 8$ mice per group. *$P < 0.05$ versus WT; #$P < 0.05$ versus WT-IR mice. $P$ values were determined by one-way ANOVA test followed by Bonferroni post hoc analysis. Data are expressed as mean ± SEM. CIRBIL-KO CIRBIL knockout mice, CIRBIL-KO-IR CIRBIL knockout mice with ischemia–reperfusion.

Fig. 6B–D). These results suggest that downregulation of lncCIRBIL is detrimental to cardiac I/R injury.

**LncCIRBIL binds to Bclaf1 and inhibits its nuclear translocation**. To investigate the mechanism underlying the effects of lncCIRBIL on cardiac injury, we performed RNA pulldown assay

to screen the proteins that can bind lncCIRBIL. The mass spectrometry (MS) technique was used to identify the proteins in the bands associated with the sense sequence of lncCIRBIL (Fig. 3A). A total of 454 proteins were identified (Supplementary Data 1) and the apoptosis-related protein Bclaf1 was found present in the MS data. Significant pulldown of Bclaf1 by the sense sequence of lncCIRBIL but not the antisense sequence was confirmed

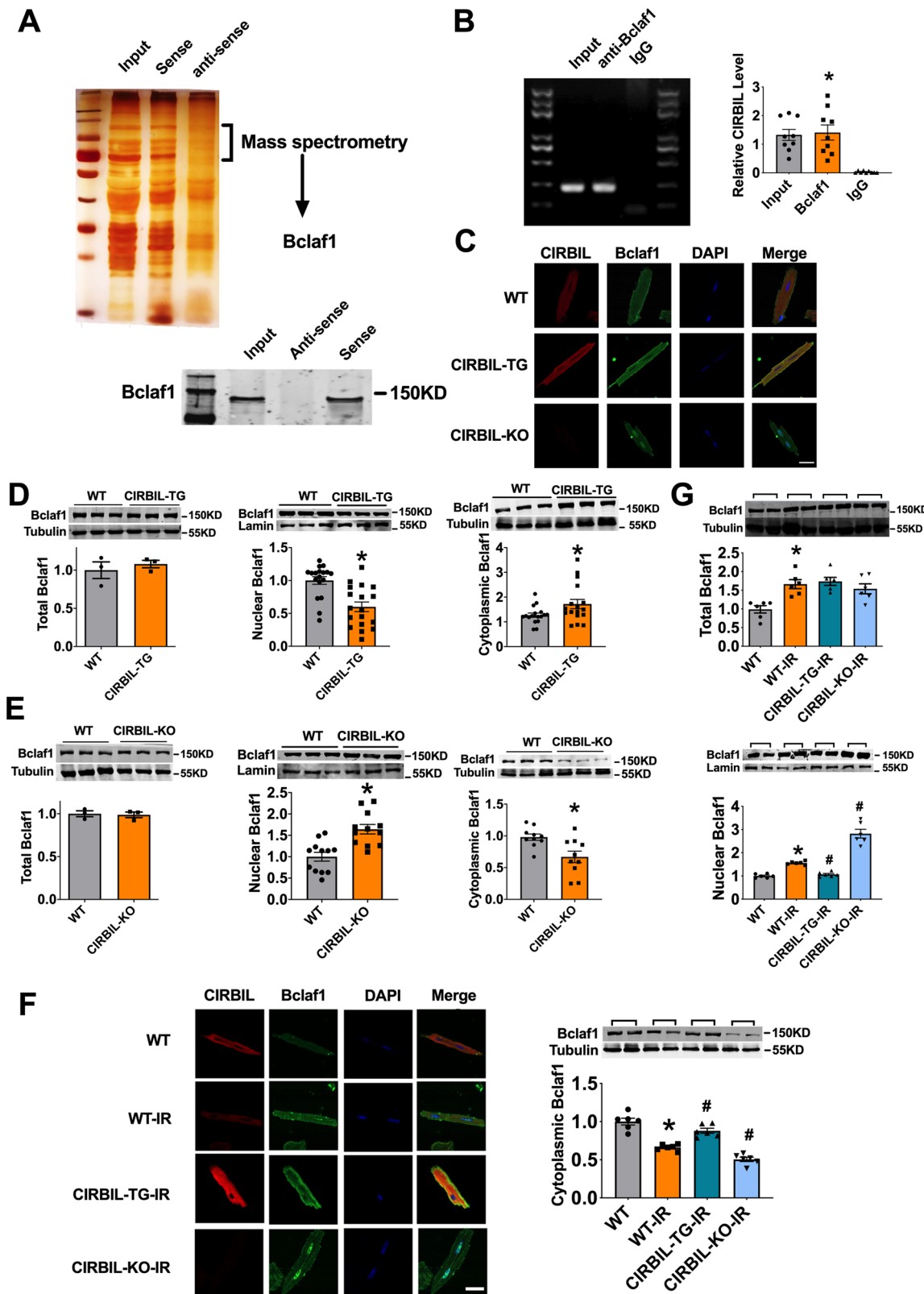

**Fig. 3 LncCIRBIL binds to Bclaf1 and inhibits its nuclear translocation. A** Upper panel: silver staining of proteins pulled down by lncCIRBIL; lower panel: western blot analysis of Bclaf1 pulled down by lncCIRBIL. The data have been reproduced in 3 independent experiments. **B** RNA immunoprecipitation (RIP) of lncCIRBIL by Bclaf1 antibody in mice heart tissue. $N = 9$ from 3 mice. $*P < 0.05$ versus IgG group. $P$ values were determined by unpaired $t$ test.
**C** LncCIRBIL distribution by in situ hybridization and Bclaf1 by immunofluorescence staining in isolated adult cardiomyocytes from WT, lncCIRBIL-TG, and lncCIRBIL-KO mice (scale bar: 20 μm). **D, E** Bclaf1 protein levels in subcellular fractions determined by western blot. Lamin is the loading control for nuclear extracts. $N = 3$ for total protein; $N = 18$ for TG, 12 for KO in nuclear protein; $N = 17$ for TG, 10 for KO in cytoplasmic protein. $*P < 0.05$ versus WT group. $P$ values were determined by unpaired $t$ test. **F** Distribution of lncCIRBIL and Bclaf1 in isolated cardiomyocytes after ischemia–reperfusion injury (scale bar: 20 μm). **G** Detection of Bclaf1 in subcellular fractions by Western blotting after I/R injury. $N = 6$. $*P < 0.05$ versus WT-IR group, $\#P < 0.05$ versus WT-IR group. $P$ values were determined by one-way ANOVA test followed by Bonferroni post hoc analysis. Data are expressed as mean ± SEM.

(Fig. 3A). Conversely, the Bclaf1 antibody precipitated a substantial amount of lncCIRBIL (Fig. 3B).

Next, we performed in situ hybridization and immunofluorescence to detect the relative subcellular distributions of lncCIRBIL and Bclaf1 in cardiomyocytes. The results showed that lncCIRBIL is located in the cytoplasm and Bclaf1 in both the cytoplasm and nucleus (Fig. 3C). Notably, such a distribution pattern of Bclaf1 was found being regulated by lncCIRBIL. In lncCIRBIL Tg mice, Bclaf1 was evenly scattered throughout the cytoplasm, while in lncCIRBIL knockout mice it became primarily restricted within the nucleus (Fig. 3C). Western blot data showed that transgenic overexpression of lncCIRBIL reduced the protein level of Bclaf1 in the nuclear fraction and simultaneously increased it in the cytoplasmic fraction (Fig. 3D). Conversely, knockout of lncCIRBIL produced the opposite changes (Fig. 3D, E). LncCIRBIL did not change the total protein level of Bclaf1. The data indicate that the binding of lncCIRBIL to Bclaf1 determines the subcellular distribution pattern of Bclaf1.

We then examined the effects of I/R injury on the relative distribution patterns of lncCIRBIL and Bclaf1. While Bclaf1 was primarily distributed in the cytoplasm in cardiomyocytes isolated from sham-operated control mice, it accumulated within the nucleus in cardiomyocytes of I/R mice. These alterations were essentially abolished by transgenic overexpression of lncCIRBIL but were exacerbated by lncCIRBIL knockout (Fig. 3F). Western blot results displayed that the total protein level of Bclaf1 was increased with I/R injury, which was not influenced by lncCIRBIL overexpression or knockout. However, the level of Bclaf1 in the nuclear fraction was increased after I/R injury, and this nucleic accumulation was mitigated by lncCIRBIL transgenic overexpression but enhanced by lncCIRBIL knockout (Fig. 3G). The alteration of Bclaf1 protein level in the cytoplasmic fraction went opposite to that of nuclear fraction (Fig. 3G). These data indicate that lncCIRBIL inhibits the nuclear accumulation but not the expression of Bclaf1 during cardiac I/R injury.

**Overexpression of Bclaf1 aggravates cardiac I/R injury.** The interaction between lncCIRBIL and Bclaf1 indicates that Bclaf1 plays an important role in regulating the effects of lncCIRBIL on cardiac injury. It has been reported that Bclaf1 overexpression induces apoptosis of cancer cells[9,10]; however, whether it also does so in the heart remained unknown. To test this note, we thus generated cardiac-specific Bclaf1 transgenic mice (Supplementary Fig. 7). The protein level of Bclaf1 was increased to twofold in Bclaf1 Tg mice relative to wild-type controls (Fig. 4A). Transgenic overexpression of Bclaf1 exaggerated the impairment of cardiac function in I/R injury mice with greater decreases in the EF and FS and increase in infarct size compared with those in WT-IR mice (Fig. 4B, C). The serum levels of LDH and CKMB were elevated in cardiac I/R injury, and Bclaf1 overexpression magnified these alterations (Fig. 4D, E). Bclaf1 overexpression also magnified the increases in caspase-3 activity, cleaved caspase-3 protein level, and TUNEL-positive cells induced by I/R injury (Fig. 4F–H). Moreover, Bclaf1 overexpression and I/R injury

elicited an additive upregulation of the expression of apoptosis-related proteins p53 and Bax, known to be the target genes of Bclaf1[11,17], at both mRNA and protein levels (Fig. 4I, J). Meanwhile, while I/R injury rendered the nuclear accumulation of Bclaf1, Bclaf1 transgenic overexpression enhanced the effect of I/R injury (Fig. 4K). These data suggest that Bclaf1 is detrimental to cardiac I/R injury.

Similar effects of Bclaf1 overexpression in H/R-induced cardiomyocyte injury were consistently observed. Transfection of Bclaf1-carrying plasmids led to a 2.2-fold increase in Bclaf1 protein level (Supplementary Fig. 8A) and this overexpression of Bclaf1 aggrandized the H/R-induced increases of caspase-3 activity, cleaved caspase-3 protein level, and TUNEL-positive cells (Supplementary Fig. 8B–D), as well as the upregulation of mRNA and protein levels of p53 and Bax (Supplementary Fig. 8E, F).

**Knockout of Bclaf1 alleviates cardiac I/R injury.** To confirm the role of Bclaf1 in cardiac I/R injury, we employed a loss-of-function approach using Bclaf1 knockout mice with adeno-associated virus 9 carrying the cas9-Bclaf1-vector (AAV9-Cas9-Bclaf1). Infection of AAV9-Cas9-Bclaf1 partially knocked out the protein expression of Bclaf1 (Fig. 5A). AAV9-Cas9-Bclaf1 recovered the I/R-induced decreases in EF and FS in mice (Fig. 5B) and reduced the infarct size and serum levels of LDH and CKMB (Fig. 5C–E). Cardiomyocyte apoptosis was inhibited by AAV9-Cas9-Bclaf1, as indicated by reduced caspase-3 activity, cleaved caspase-3 protein level, and TUNEL-positive cells (Fig. 5F–H). In addition, AAV9-Cas9-Bclaf1 suppressed the I/R-induced aberrant upregulation of p53 and Bax at both the mRNA and protein levels (Fig. 5I, J).

We next repeated the above experiments in vitro with cultured cardiomyocytes employing the siRNA to knock down endogenous Bclaf1 (Supplementary Fig. 9A). Consistent with the in vivo results, Bclaf1 siRNA diminished the H/R-induced increase in the caspase-3 activity, cleaved caspase-3 protein level, and TUNEL-positive cells (Supplementary Fig. 9B–D). Moreover, it also repressed the abnormal upregulation of p53 and Bax during H/R (Supplementary Fig. 9E, F). These data imply that knockdown of Bclaf1, like overexpression of lncCIRBIL, is protective against cardiac injury.

**Bclaf1 mediates the regulation of lncCIRBIL on cardiomyocyte injury induced by H/R.** We then turned to get insight into the mechanistic link between lncCIRBIL and Bclaf1 to see whether the former acts by targeting the latter to manifest its cardioprotective effects in the settings of I/R and H/R injuries. The cultured cardiomyocytes were co-transfected with the plasmids for lncCIRBIL and Bclaf1 overexpression, followed by H/R procedures. Transfection of lncCIRBIL-overexpressing plasmid alone protect the cardiomyocytes from H/R injury, but the beneficial effect was countered by co-transfection with Bclaf1 plasmid that resulted in substantial increases in the caspase-3 activity, cleaved caspase-3 protein level, and TUNEL-positive cells (Fig. 6A–C). Meanwhile, co-transfection of Bclaf1 abrogated the suppressive

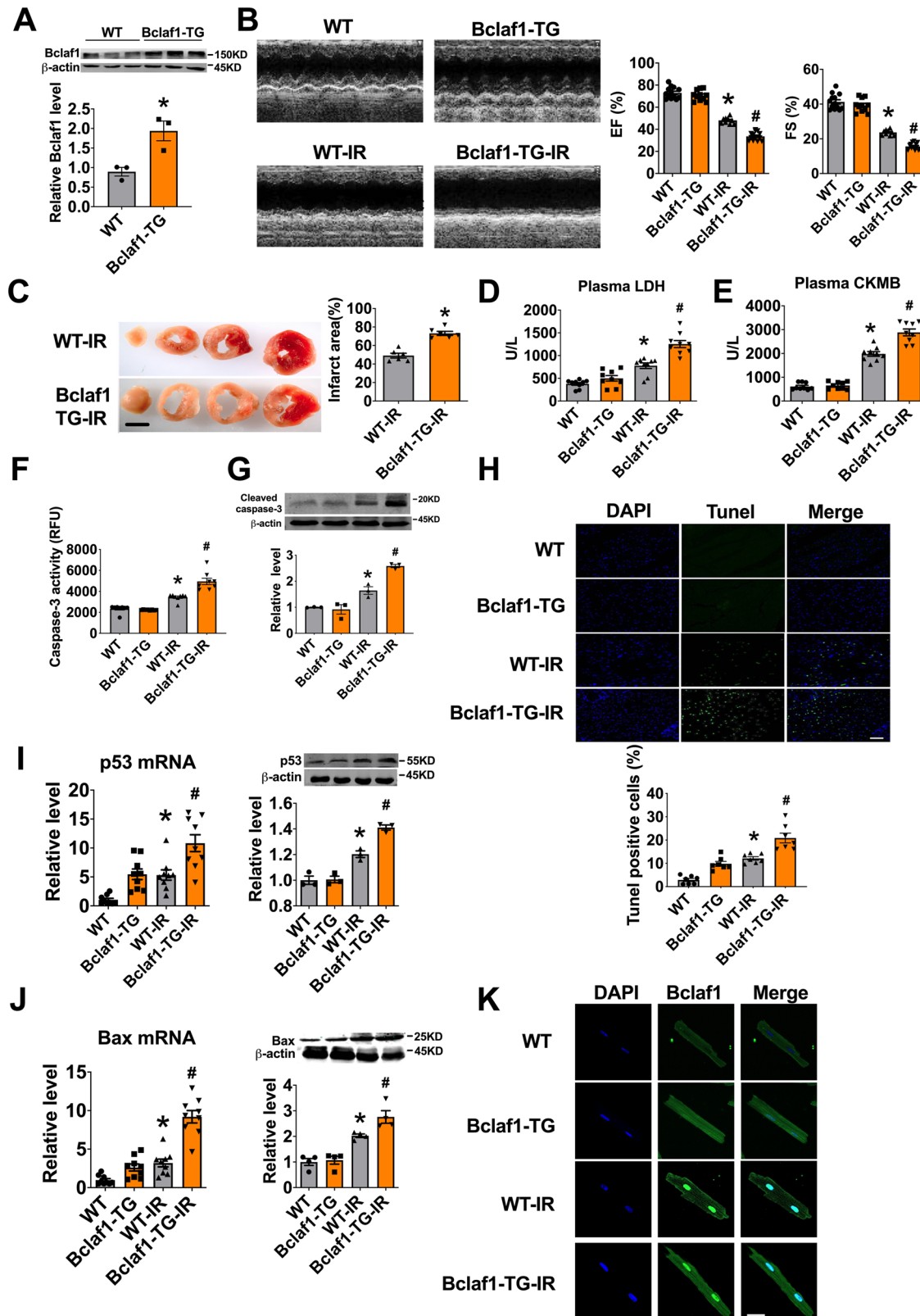

effects on p53 and Bax expression elicited by lncCIRBIL over-expression alone (Fig. 6D, E). Conversely, the pro-apoptotic effects of lncCIRBIL siRNA were counteracted by Bclaf1 siRNA, as manifested by the reduced caspase-3 activity, cleaved caspase-3 protein level, and TUNEL-positive cells in H/R cardiomyocytes (Fig. 6F–H). Furthermore, the upregulation of mRNA and pro-tein expression of p53 and Bax seen with lncCIRBIL siRNA alone

was prevented by co-transfection of Bclaf1 siRNA (Fig. 6I, J). Thus the protective effects of lncCIRBIL on cardiac injury are highly attributable to the functional deficiency of Bclaf1.

**Partial knockout of Bclaf1 abrogates the exacerbating effects of lncCIRBIL knockout on cardiac I/R injury.** To generate more evidence for the role of Bclaf1 in mediating the effects of

**Fig. 4 Cardiomyocyte-specific transgenic overexpression of Bclaf1 aggravates cardiac ischemia–reperfusion injury. A** Bclaf1 protein level determined by western blot. $N = 3$. *$P < 0.05$ versus WT group. $P$ values were determined by unpaired $t$ test. **B** Representative images of echocardiographs and statistics of ejection fraction (EF) and fractional shortening (FS). WT $N = 13$, Bclaf1-TG $N = 11$, WT-I/R $N = 10$, Bclaf1-TG-I/R $N = 13$. *$P < 0.05$ versus WT, #$P < 0.05$ versus WT-IR mice. $P$ values were determined by one-way ANOVA test followed by Bonferroni post hoc analysis. **C** Infarct area determined by TTC staining (scale bar: 2 mm). $N = 6$ mice per group. *$P < 0.05$ versus WT-IR mice. $P$ values were determined by unpaired $t$ test. **D, E** Serum concentrations of LDH and CKMB, respectively, measured by Elisa assay. $N = 9$ mice per group. *$P < 0.05$ versus WT, #$P < 0.05$ versus WT-IR mice. $P$ values were determined by one-way ANOVA test followed by Bonferroni post hoc analysis. **F, G** Caspase-3 activity and cleaved caspase-3 protein level, respectively. $N = 8$ mice per group for caspase activity, and $N = 3$ mice for cleaved caspase-3 protein level. *$P < 0.05$ versus WT, #$P < 0.05$ versus WT-IR mice. $P$ values were determined by one-way ANOVA test followed by Bonferroni post hoc analysis. **H** Apoptosis of cardiomyocyte in the border zone (BZ) by TUNEL staining (scale bar: 20 μm). $N = 7$ mice per group. *$P < 0.05$ versus WT, #$P < 0.05$ versus WT-IR mice. $P$ values were determined by one-way ANOVA test followed by Bonferroni post hoc analysis. **I** mRNA and protein levels of p53 by RT-PCR and western blot. $N = 9$ for PCR, $N = 3$ for western blot. *$P < 0.05$ versus WT, #$P < 0.05$ versus WT-IR mice. $P$ values were determined by one-way ANOVA test followed by Bonferroni post hoc analysis. **J** mRNA and protein levels of Bax by RT-PCR and western blot. $N = 9$ for PCR, $N = 4$ for western blot. *$P < 0.05$ versus WT, #$P < 0.05$ versus WT-IR mice. $P$ values were determined by one-way ANOVA test followed by Bonferroni post hoc analysis. **K** Distribution of Bclaf1 in isolated cardiomyocytes (scale bar: 20 μm). Data are expressed as mean ± SEM.

lncCIRBIL on cardiac I/R injury, we went on to investigate the influence of Bclaf1 knockdown on cardiac injury in lncCIRBIL knockout mice. We partially knocked out the expression of Bclaf1 in lncCIRBIL knockout mice by injecting the AAV9-Cas9-Bclaf1 and subjected the mice to cardiac I/R injury. We found that knockout of lncCIRBIL significantly impaired cardiac function in I/R injury and the harmful effect was mitigated by partial deletion of Bclaf1 (Fig. 7A). Meanwhile, partial deletion of Bclaf1 reversed the increase in infarct size, caspase-3 activity, cleaved caspase-3 protein level, and TUNEL-positive cells in lncCIRBIL-deficient mice undergoing cardiac I/R injury (Fig. 7B–E). The increase of p53 and Bax expression in lncCIRBIL knockout mice undergoing I/R was prevented by partial knockout of Bclaf1 (Fig. 7F). These data suggest that Bclaf1 is a downstream molecule that mediates the effects of lncCIRBIL on cardiac I/R injury.

**The expression of lncCIRBIL is regulated by p53.** LncCIRBIL is an intergenic lncRNA and it likely contains the sequence elements for transcriptional regulation. To test this notion, we analyzed the sequence using PROMO database (http://alggen.lsi.upc.es/cgi-bin/promo_v3/promo/promoinit.cgi?dirDB=TF_8.3) to predict the potential binding motifs for transcriptional factors that can regulate the expression of lncCIRBIL. We identified p53 as a candidate transcriptional factor of lncCIRBIL expression. We then experimentally validated p53 as a transcriptional repressor of lncCIRBIL expression. First, overexpression of p53 in cultured cardiomyocytes reduced the level of lncCIRBIL transcripts (Supplementary Fig. 10A). Knockdown of p53 did not significantly change the expression of lncCIRBIL under normoxic culture conditions (Supplementary Fig. 10B). However, while the cells were exposed to H/R, knockdown of p53 pronouncedly increased lncCIRBIL transcripts (Supplementary Fig. 10C). These data indicate that lncCIRBIL is negatively regulated by p53.

On the other hand, we observed that the downregulation of lncCIRBIL in WT-IR mice was partially restored by Bclaf1 knockout (Supplementary Fig. 10D).

**Discussion**
In this study, we found that lncCIRBIL was downregulated during cardiac I/R injury and it produced cardioprotective role against cardiac injury by suppressing cardiomyocyte apoptosis and reducing infarct size to improve overall cardiac function. The evidence supports that lncCIRBIL acts by direct binding to Bclaf1, a detrimental molecule in cardiac I/R injury, to block its nuclear translocation and p53 activation. LncCIRBIL deficiency alone produced I/R-like phenotypes and partial knockout of Bclaf1 prevented this deleterious effect. The expression of lncCIRBIL is

negatively regulated by p53 at the transcriptional level. These findings indicate that lncCIRBIL exerts cardioprotective effects by inhibiting the nuclear translocation of Bclaf1 and the subsequent activation of p53 transcription. These findings prompted us to propose a positive feedback regulatory circuit leading to worsening I/R or H/R injuries: I/R or H/R → lncCIRBIL↓ → Bclaf1↑ → p53↑ → lncCIRBIL↓ → I/R or H/R injury↑ (Fig. 8).

LncRNAs are critical in cardiac development and various pathological processes, such as coronary artery disease, myocardial infarction, and heart failure[19]. The specific role of lncRNAs in cardiac ischemia injury has been the focus in the field of cardiovascular research. The alteration in lncRNA expression is well connected to development and progression of various diseases. Liu et al. showed that 64 lncRNAs are upregulated and 87 downregulated in I/R hearts of mice[20]. Several dysregulated lncRNAs have been functionally validated as important regulators of cardiac injury. LncRNA CARL inhibits anoxia-induced cardiomyocyte apoptosis by impairing miR-539-dependent prohibitin 2 downregulation[21]. LncRNA autophagy-promoting factor (APF) regulates autophagy and myocardial infarction by targeting miR-188-3p[22]. LncRNA necrosis-related factor (NRF) regulates programmed necrosis and myocardial injury during ischemia and reperfusion by targeting miR-873[23]. LncRNA H19 directly binds to miR-103/107 and regulates cardiomyocyte necrosis[24]. Our previous study showed that lncRNA H19 alleviated cardiac I/R injury by inhibiting miR-877-3p-mediated downregulation of Bcl-2[25]. In this study, we screened for the dysregulated lncRNAs in I/R hearts of mice and identified the downregulated lncCIRBIL as a key regulatory lncRNA of cardiac injury. LncCIRBIL was significantly downregulated in the infarct zone and the border of infarct area compared with remote NIZ. The data from lncCIRBIL transgenic overexpression and knockout mice proved that lncCIRBIL is protective against cardiac I/R injury. These findings thus provided substantial evidence supporting the beneficial role of lncCIRBIL in cardiac I/R injury.

LncRNAs exhibit their effects through diverse mechanisms, including epigenetic regulation, gene transcription, protein stability, and compartmentalization by binding to proteins that mediate these functions[4]. The specific regulatory mechanism of a lncRNA is closely related its cellular localization. LncRNAs residing in the nucleus tend to regulate gene expression in *cis* or *trans* by recruiting chromatin modifiers or transcription factors, while lncRNAs in the cytoplasm may modulate translation, act as competing endogenous RNAs, and regulate protein modifications and trafficking. For example, lncRNA X-linked X-inactive-specific transcript inhibits gene transcription by recruiting polycomb repressive complex 2 to the X chromosome[26]. LncRNA H19 induces repressive histone modification in the target genes

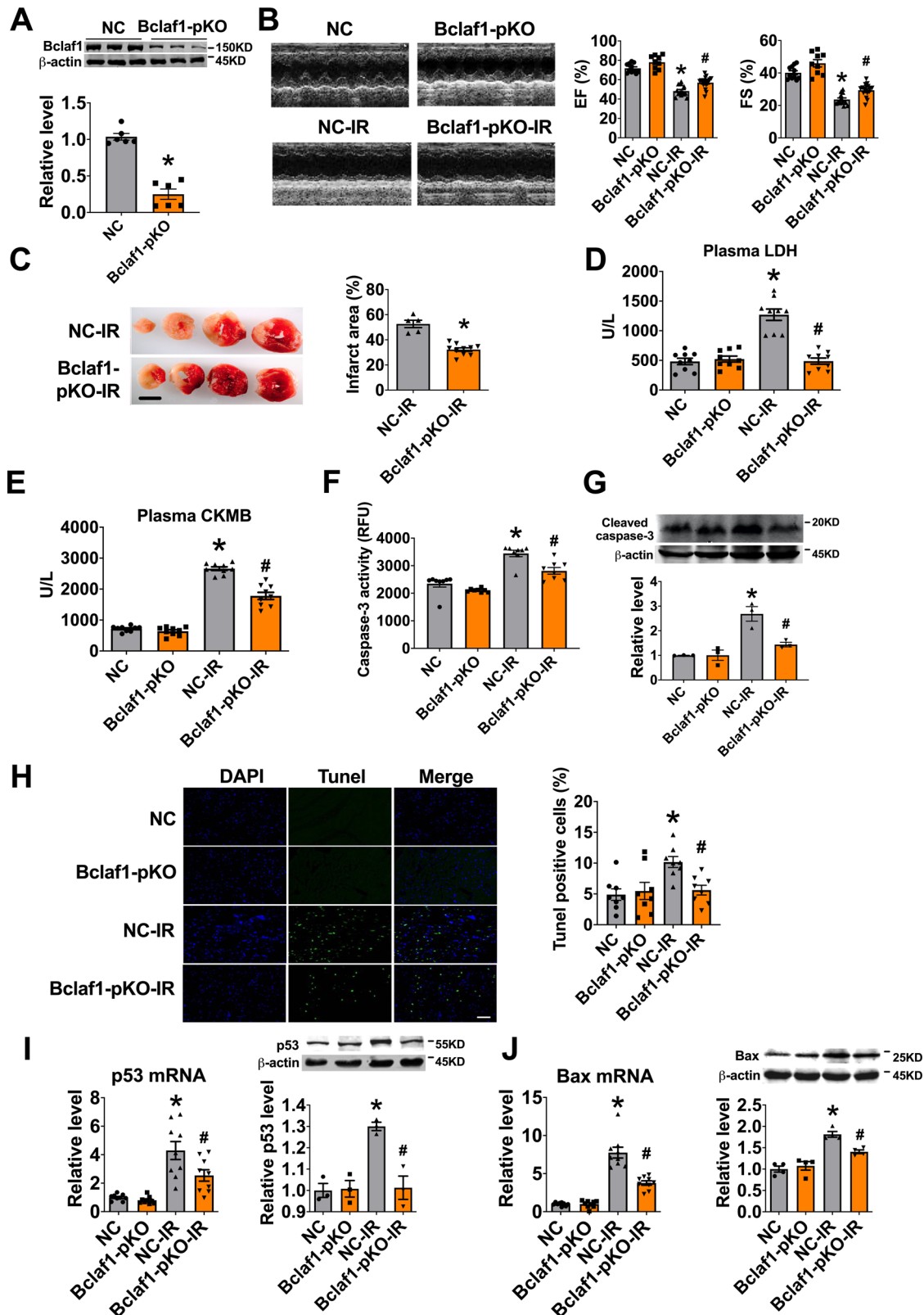

and finally controls target gene expression by recruiting Methyl-CpG-binding domain protein 1[27]. LncRNA HOTAIR mediates the interaction between Snail and enhancer of zeste homolog 2 to repress downstream targets. We previously found that lncRNA CCRR improves cardiac conduction by blocking endocytic trafficking of connexin43 (Cx43) through binding to Cx43 interacting protein CIP85[28]. In dissecting the mechanism for the effects

of lncCIRBIL in cardiac I/R injury, we first examined its cellular distribution and discovered that it was restricted in the cytoplasm. Then we screened for the proteins that may bind lncCIRBIL and mediate its effects. The RNA pulldown and RNA immunoprecipitation (RIP) assay reciprocally proved that lncCIRBIL directly binds to Bclaf1. We further confirmed the effect of lncCIRBIL was mediated by Bclaf1. In terms of the reported lncRNAs in cardiac

**Fig. 5 Partial knockout (pKO) of Bclaf1 alleviates cardiac ischemia–reperfusion injury. A** Bclaf1 in the heart tissue after adeno-associated virus carrying cas9-Bclaf1 injection. $N = 6$. *$P < 0.05$ versus NC group. $P$ values were determined by unpaired $t$ test. **B** Representative images of echocardiographs and statistics of ejection fraction (EF) and fractional shortening (FS). NC $N = 11$, Bclaf1-pKO $N = 9$, NC-I/R $N = 13$, Bclaf1-pKO-I/R $N = 18$. *$P < 0.05$ versus NC, #$P < 0.05$ NC-IR. $P$ values were determined by one-way ANOVA test followed by Bonferroni post hoc analysis. **C** Infarct size by TTC staining (scale bar: 2 mm). NC-I/R $N = 5$, Bclaf1-pKO-I/R $N = 10$. *$P < 0.05$ versus NC-I/R, $P$ values were determined by unpaired $t$ test. **D, E** Serum concentrations of LDH and CKMB, respectively, measured by Elisa assay. $N = 9$ mice per group. *$P < 0.05$ versus NC, #$P < 0.05$ NC-IR. $P$ values were determined by one-way ANOVA test followed by Bonferroni post hoc analysis. **F** Caspase-3 activity. $N = 8$ mice per group. *$P < 0.05$ versus NC, #$P < 0.05$ NC-IR. $P$ values were determined by one-way ANOVA test followed by Bonferroni post hoc analysis. **G** Cleaved caspase-3 protein level. $N = 3$ mice per group. *$P < 0.05$ versus NC, #$P < 0.05$ NC-IR. $P$ values were determined by one-way ANOVA test followed by Bonferroni post hoc analysis. **H** Apoptosis of cardiomyocyte in the border zone (BZ) by TUNEL staining (scale bar: 20 μm). $N = 8$ mice per group. *$P < 0.05$ versus NC, #$P < 0.05$ NC-IR. $P$ values were determined by one-way ANOVA test followed by Bonferroni post hoc analysis. **I** The mRNA and protein levels of p53, respectively. $N = 9$ for mRNA and $N = 3$ for protein. *$P < 0.05$ versus NC, #$P < 0.05$ NC-IR. $P$ values were determined by one-way ANOVA test followed by Bonferroni post hoc analysis. **J** The mRNA and protein levels of Bax, respectively. $N = 9$ for mRNA and $N = 4$ for protein. *$P < 0.05$ versus NC, #$P < 0.05$ NC-IR. $P$ values were determined by one-way ANOVA test followed by Bonferroni post hoc analysis. Data are expressed as mean ± SEM.

injury, most of them (CARL, APF, NRF, H19) exhibit their effects by sponging their respective target miRNAs that are related to cardiac injury[22,23,25,29]. Few studies have explored the interaction between lncRNA and proteins as the mechanism of actions in cardiac ischemia. To our knowledge, the only one example is lncRNA CAIF, which was shown to attenuate myocardial infarction by binding p53 and preventing the transcription of its target gene myocardin[30].

Bclaf1 was originally discovered as a protein that interacts with Bcl2, which is mainly located in the nucleus and can be sequestered to the cytoplasm by Bcl2[9]. Subsequent studies showed that Bclaf1 can induce apoptosis by directly activate the transcription of p53[9–11]. In cultured colon adenocarcinoma cells, Bclaf1 overexpression increases the levels of pro-apoptotic proteins, p53 and Bax[17]. Consistently, our results verified Bclaf1 as a detrimental molecule in cardiac I/R injury. Bclaf1 upregulated the expression of p53 and Bax, and the level of Bclaf1 in the nucleus increased during cardiac I/R injury, indicating that Bclaf1 translocation to the nucleus to activate p53 and Bax expression is a necessary step leading to cardiac I/R injury. This is the first evidence for the function of Bclaf1 and underlying mechanism in cardiac I/R injury.

Additionally, upon binding to Bclaf1, lncCIRBIL altered the cellular distribution of Bclaf1. Overexpression of lncCIRBIL restricted Bclaf1 in the cytoplasm, while the knockout of lncCIRBIL retained it to the nucleus. This is another piece of evidence for the nuclear translocation of Bclaf1 as a critical process in the induction of cardiac I/R injury due to down-regulation of lncCIRBIL contributes the pathogenesis of cardiac I/R injury. This notion was confirmed by the rescuing experiments that partial knockout of Bclaf1 prevented the harmful effects of lncCIRBIL knockout on cardiac I/R injury. Meanwhile, lncCIR-BIL did not change the protein expression of Bclaf1, indicating that the binding of lncCIRBIL to Bclaf1 does not affect the translation or stability of Bclaf1.

Collectively, the present study revealed that lncCIRBIL and Bclaf1 are critical regulators of cardiac I/R injury. LncCIRBIL exerts its beneficial effects by sequestering Bclaf1 in the cytoplasm and preventing it from nuclear translocation to activate the p53 transcriptional activity in the nucleus. It is therefore speculated that both lncCIRBIL and Bclaf1 hold the potential as therapeutic targets for the development of cardioprotective agents.

## Methods

**Human plasma samples**. Human plasma samples were collected from the Department of Cardiology, the First Affiliated Hospital of Harbin Medical University (Harbin, China) between June 2020 and September 2020 from 32 AMI patients and 32 non-AMI control subjects. AMI was diagnosed based on combination of several parameters: ischemic symptoms plus increased cardiac troponin I and CKMB, appearance of pathological Q wave, and ST-segment elevation or depression defined by the European Society of Cardiology/American College of

Cardiology. The use of human samples was approved by the institutional review board of the Harbin Medical University and conformed to the Declaration of Helsinki. Written informed consent was obtained from each of the participants.

**Animals**. Male adult mice (8 weeks old) were used in the present study. Mice were housed in a facility with 12-h light/12-h dark cycle at $23 \pm 3$ °C and 30–70% humidity. All experiments complied with the guiding principles for the care and use of laboratory animals in Harbin Medical University and were approved by the Ethics Committee for Animal Experimentation of School of Pharmacy, Harbin Medical University.

**Mouse model of cardiac I/R injury**. Mice were anesthetized with 2% avertin (0.1 mL/10 g body weight). I/R injury in mice was induced by 45 min ischemia, followed by 24 h reperfusion. After 24-h reperfusion, the heart was rapidly excised and sectioned serially into 1–2-mm-thick sections. Then the slices were incubated in 2.0% 2,3,5-triphenyltetrazolium chloride (Solarbio, Beijing, China) at 37 °C for 15 min for measurement of the infarct area. The staining was stopped by ice-cold sterile saline and the slices were fixed in 4% neutral buffered formaldehyde for another 5 min. Both sides of each slice were digitally photographed by using somatic microscope (Zeiss Stemi 508, Jena, Germany). The infarct area was measured and determined using computerized planimetry (Image pro-plus 6.0).

**Mouse neonatal cardiomyocyte isolation and culture**. Cardiomyocytes were isolated from 1- to 2-day-old mice. Briefly, the isolated hearts were washed and minced in Dulbecco's modified Eagle's medium (DMEM) buffer (Biological Industries, Haemek, Israel). The cardiac specimens were then dispersed in pancreatin (Beyotime, Shanghai, China), and the lysates were collected. After centrifugation at $1500 \times g$ for 5 min, the precipitated cells were resuspended in DMEM buffer supplemented with 5% fetal bovine serum (Biological Industries, Haemek, Israel), and 0.8% penicillin and streptomycin (Beyotime, Shanghai, China). The cells were pre-plated and cultured in humidified 5% $CO_2$ incubator at 37 °C for 2 h. The dissociated cardiomyocytes were then collected and plated for another 48 h before use for subsequent experiments. The cultured cardiomyocytes were incubated under hypoxic condition (5% $CO_2$ and 95% $N_2$) for 12 h, followed with reoxygenation (H/R) for 24 h to establish in vitro H/R model.

**Generation of lncCIRBIL transgenic and knockout mice**. LncCIRBIL cardiac-specific transgenic overexpression and global knockout mice were generated by Cyagen Biosciences Inc (Guangzhou, China). Briefly, the sequence of lncCIRBIL was cloned into the murine α-MHC promoter (mouse α-cardiac myosin heavy chain promoter, a cardiomyocyte-specific expression promoter) expression vector and the obtained DNA fragment containing lncCIRBIL driven by α-MHC promoter was microinjected into the fertilized eggs. The genomic DNA was prepared from tail tissue and subjected to PCR amplification to identify lncCIRBIL transgenic offspring. The primers used for verifying transgenic gene in mice are: forward 5′-GAAGTGGTGGTGTAGGAAAGTCTAG-3′ and reverse 5′-CCCCATCCCTG CAGGCATTC-3′.

The lncCIRBIL knockout mouse line was generated by the CRISPR/Cas-mediated genome engineering. In brief, two exons were identified and exon2 was selected as the target site. Cas9 mRNA and gRNA generated by in vitro transcription were then injected into the fertilized eggs for lncCIRBIL knockout productions. The founders were genotyped by PCR followed by DNA sequencing analysis, and the positive founders were bred to the next generation that was further verified by PCR genotyping and DNA sequencing analysis. To genotype offspring, genomic PCR of tail DNA was performed with forward 5′-TGCCATA TCACCATATGCACAACT-3′ and reverse 5′-TGTCAAAGTCATCAGTACTTTA TGGGTT-3′ primers. All mice were compared only to non-transgenic or wild-type gender-matched littermates and were about 8 weeks old.

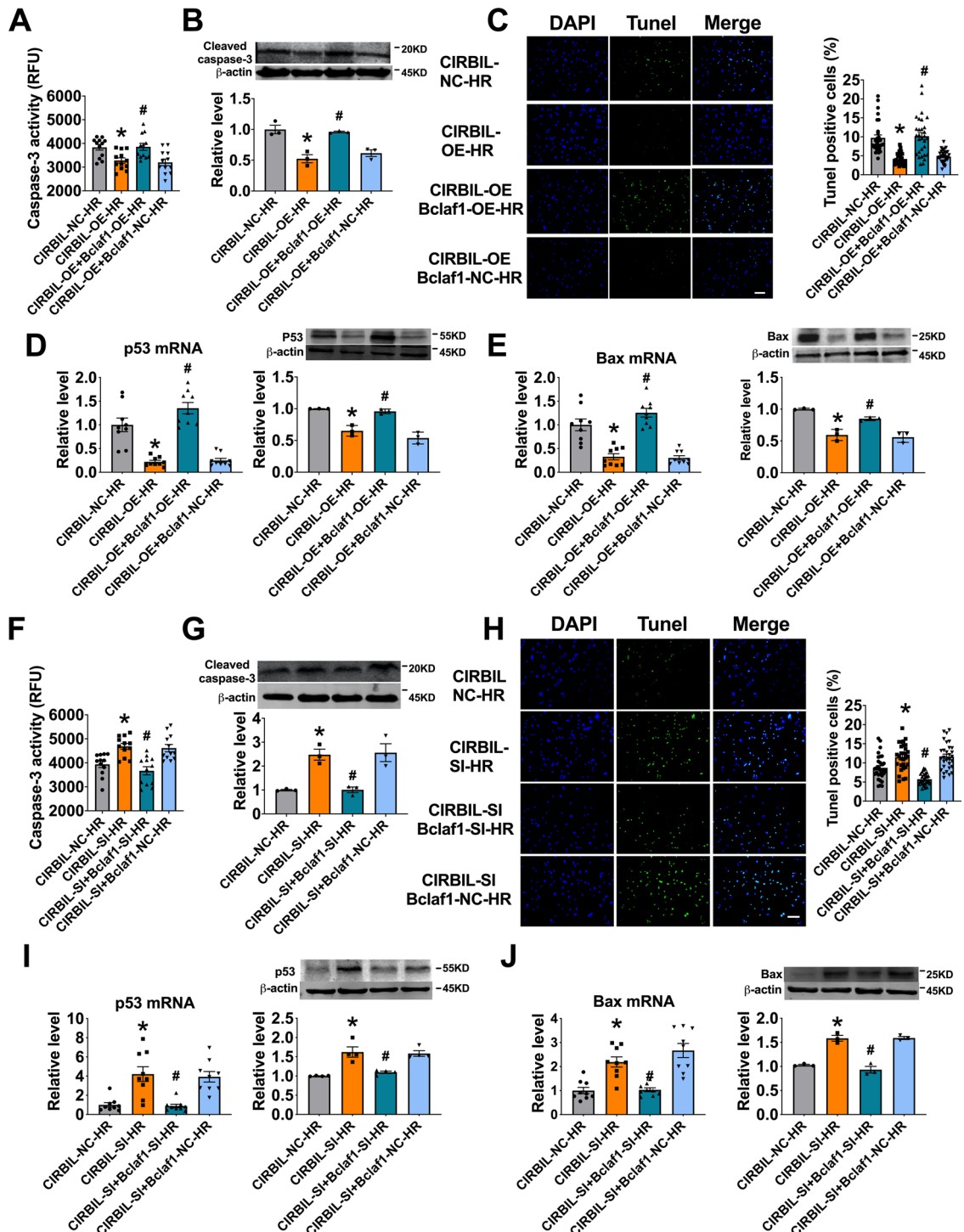

**Generation of mouse line with cardiomyocyte-specific Bclaf1 overexpression.** To analyze the function of Bclaf1 in cardiomyocytes in vivo, cardiomyocyte-specific Bclaf1 transgenic overexpression mice were generated. A transgene containing Bclaf1 was cloned downstream of the α-MHC promoter. A fragment containing the promoter region and transgene sequence was agarose gel purified and micro-injected into the pronucleus of one-cell mouse embryos of C57BL/6 mice at Cyagen Biosciences Inc. (Guangzhou, China). Transgenic mice were identified by conventional PCR of genomic DNA from mouse tissue, and the relevant primer pair used for verifying the transgenic mice is: forward 5′-AGAGCCA-TAGGCTACGGTGTA-3′, reverse 5′-TCTGTACCCATAAGGTCGTCTC-3′. Control mice were C57BL/6 mice or negative littermate mice. The mice used for our experiments were all male at an age of 8 weeks.

**Construction of Bclaf1 knockdown virus.** The recombinant serum type 9 adeno-associated virus system (AAV9) was used to construct CRISPR/Cas9-mediated

mouse bclaf1 gene (Gene ID: 72567) knockdown vector (Cyagen Biosciences, Guangzhou, China). For CRISPR/Cas9-mediated bclaf1 knockdown vector, we designed two bclaf1 gRNA targets (bclaf1 gRNA#1 5′-GTCCTAGACGAGGTCGG TCA-3′ and bclaf1 gRNA#2 5′-TGGCAGGGCTTCTTCGTGAT-3′), synthesized an artificial DNA fragment with Bbs I restriction enzyme site at each terminus, and digested both the pAAV[gRNA]-mCherry-U6> shuttle empty vector and the artificial DNA fragment with Bbs I. Finally, this artificial DNA fragment was cloned into pAAV[gRNA]-mCherry-U6>Shuttle empty vector to produce pAAV [2gRNA]-mCherry-U6>mBclaf1 gRNA. The virus at a dose of $1 \times 10^{10}$ genome copies per animal were injected into 6-week-old C57BL/6 mice via tail vein. Two weeks after injection, experiment interventions were performed.

**Echocardiography.** Mice were anesthetized by intraperitoneal injection of 2% avertin (0.1 mL/10 g body weight) and examined by M-mode echocardiography using a Vevo2100 Imaging System (VisualSonics, Toronto, Canada) at 24 h post-operation.

**Fig. 6 Bclaf1 mediates the regulation of lncCIRBIL on cardiomyocyte injury induced by hypoxia/reoxygenation. A, B** Caspase-3 activity and cleaved caspase-3 protein level in cardiomyocytes co-transfected with lncCIRBIL- and Bclaf1-overexpressing plasmids. $N = 12$ from 3 independent cultures for caspase-3 activity, $N = 3$ for cleaved caspase-3 protein level. NC negative control (empty plasmid), OE lncCIRBIL overexpression plasmids. *$P < 0.05$ versus lncCIRBIL-NC-HR. #$P < 0.05$ versus lncCIRBIL-OE-HR group. $P$ values were determined by one-way ANOVA test followed by Bonferroni post hoc analysis. **C** Cardiomyocyte apoptosis by TUNEL staining (scale bar: 20 μm). $N = 10$, from 3 independent cultures. NC negative control (empty plasmid), OE lncCIRBIL overexpression plasmids. *$P < 0.05$ versus lncCIRBIL-NC-HR, #$P < 0.05$ versus lncCIRBIL-OE-HR. $P$ values were determined by one-way ANOVA test followed by Bonferroni post hoc analysis. **D, E** The mRNA and protein expression of p53 and Bax by qRT-PCR and western blot, respectively. $N = 9$ for qRT-PCR, and $N = 3$ for western blot. NC negative control (empty plasmid), OE lncCIRBIL overexpression plasmid. *$P < 0.05$ versus lncCIRBIL-NC-HR, #$P < 0.05$ versus lncCIRBIL-OE-HR. $P$ values were determined by one-way ANOVA test followed by Bonferroni post hoc analysis. **F, G** Caspase-3 activity and cleaved caspase-3 protein level in cardiomyocytes co-transfected with lncCIRBIL siRNA and Bclaf1 siRNA. $N = 12$ from 3 independent cultures for caspase-3 activity, $N = 3$ for cleaved caspase-3 protein level. NC negative control. *$P < 0.05$ versus lncCIRBIL-NC-HR, #$P < 0.05$ versus lncCIRBIL-SI-HR. $P$ values were determined by one-way ANOVA test followed by Bonferroni post hoc analysis. **H** Cardiomyocyte apoptosis by TUNEL staining (scale bar: 20 μm). $N = 10$ form 3 independent cultures. NC negative control. *$P < 0.05$ versus lncCIRBIL-NC-HR, #$P < 0.05$ versus lncCIRBIL-SI-HR. $P$ values were determined by one-way ANOVA test followed by Bonferroni post hoc analysis. **I, J** The mRNA and protein expression of p53 and Bax by qRT-PCR and western blot, respectively. $N = 9$ for qRT-PCR, and $N = 3$ for western blot, from 3 independent batches of cells. NC negative control. *$P < 0.05$ versus lncCIRBIL-NC-HR, #$P < 0.05$ versus lncCIRBIL-SI-HR. $P$ values were determined by one-way ANOVA test followed by Bonferroni post hoc analysis. Data are expressed as mean ± SEM.

M-mode tracings of the left ventricle (LV) were acquired using the short-axis view, with the ultrasound beam perpendicular to the LV at the midpapillary level to determine EF, FS, wall thickness, LV inner diameter, and LV volume. LV dimension was averaged from a minimum of six consecutive cardiac cycles per heart.

**Adult cardiomyocyte isolation**. Mice were anesthetized by intraperitoneal injection of 2% avertin (0.1 mL/10 g body weight). Hearts were rapidly excised, and the aorta was cannulated on a constant-flow Langendorff apparatus. The heart was digested by perfusion with Tyrode's solution containing Type II collagenase (1 mg/mL), protease (0.02 mg/mL), and bovine serum albumin (BSA; 1 mg/mL). Tyrode's solution contained (mM): NaCl 123, KCl 5.4, HEPES 10, NaH$_2$PO$_4$ 0.33, MgCl$_2$ 1.0, and glucose 10; pH adjusted to 7.4 with NaOH. After the tissue was softened, LV was dissected and gently minced into small chunks, which were then equilibrated in Tyrode's solution with 200 μM CaCl$_2$ and 1% BSA. Single rod-shaped cells with clear cross-striations were used for experiments. All solutions were gassed with 95% O$_2$ and 5% CO$_2$ and warmed to 37 ± 0.5 °C.

**Cell transfection with plasmids or siRNA**. Transfection of plasmids or siRNA was carried out with lipofectamine 2000 reagent (Invitrogen, Carlsbad, America) and X-treme gene siRNA transfection reagent (Roche, Basel, Switzerland), respectively. The sequences of siRNAs for mouse lncCIRBIL were: 5′-GUAAGU GUUUAACAGUCCUTT-3′, and 5′-AGGACUGUUAAACACUUACTT-3′, for mouse Bclaf1: 5′-GAAGGACCCAAGUACAAGUTT-3′, and 5′-ACUUGUACU UGGGUCCUUCTT-3′, for mouse p53: 5′-CAUUUUCAGGCUUAUGGAATT-3′, and 5′-UUCCAUAAGCCUGAAAAUGTT-3′. Forty-eight hours after transfection, cardiomyocyte H/R model was established.

**Real-time quantitative reverse transcriptase-PCR**. Total RNA was extracted from collected cells or tissues by using Trizol reagent (Invitrogen, Carlsbad, America). DNase-treated RNA was reverse transcribed using the Trans-Script All-in-one First-strand cDNA Synthesis Supermix for qPCR Kit (TransGen Biotech, Beijing, China). Real-time quantitative PCR was performed using SYBR Green Master (Roche, Basel, Switzerland). The relative expression levels were calculated based on Ct values and were normalized to the level of β-actin as an internal control (Supplementary Table 1).

**Nuclear and cytoplasmic protein extraction**. The cytoplasmic and nuclear extracts were separated and prepared from cardiac tissue by using NE-PER nuclear and cytoplasmic extraction reagents (Thermo Fisher Scientific, New York, CN). The tissues were cut into small pieces and placed in new microcentrifuge tubes. Then 100 μL of CRE I (cytoplasmic extraction reagent I) was added and the tissues were grinded into homogenate. Then the ice-cold CRE II (cytoplasmic extraction reagent II) was added, and the sample was vortexed and centrifuged. The supernatant (cytoplasmic extract) was immediately transferred to a clean tube and the sediment was suspended with nuclear extraction reagent. The suspended sediment was vortexed and centrifuged, and the supernatant was obtained as nuclear extract.

**Western blot**. The total protein samples were purified from cells and tissues with RIPA lysis buffer (Beyotime, Beijing, China) and a protease inhibitor cocktail (Roche, Basel, Switzerland) at 4 °C. Proteins were separated by sodium dodecyl sulfate–polyacrylamide gel electrophoresis (SDS-PAGE) and transferred to polyvinylidene difluoride membranes (Pall Corporation, Mexico, USA). The membranes were blocked in Tris-buffered saline containing 5% milk and then incubated with primary antibodies at 4 °C overnight. β-Actin was used as an internal control

(Supplementary Table 2). After washing with PBST (phosphate-buffered saline (PBS) with Tween), the membranes were incubated with the secondary antibody at room temperature for 60 min. The membranes were scanned and analyzed by ODYSSEY machine (LI-COR, American).

**Immunofluorescence**. Isolated cardiomyocytes were attached to the adhesive slide. The cells were rinsed with cold PBS three times and fixed in 4% paraformaldehyde for 15 min. After rinsing with PBS three times, the cells were incubated with 0.5% Triton X-100 at room temperature for 1 h and then blocked with 10% normal goat serum at 37 °C for 1 h. Next, Bclaf1 antibody (1:500) was added into each slide and incubated at 4 °C overnight. After three rinses with PBS, the fluorescein-488 antibody (1:500) was added and the samples were incubated in a wet box at room temperature for 1 h. After rinsing with PBS three times, the nuclei were counterstained with 4,6-diamidino-2-phenylindole (DAPI) at room temperature for 15 min. Immunofluorescence was analyzed under a confocal fluorescence microscope (Carl Zeiss, Germany).

**Fluorescence in situ hybridization (FISH)**. The distribution of lncCIRBIL in cardiac myocyte was examined using the FISH Kit (Ribo, Suzhou, China). The cells were fixed by 4% paraformaldehyde and penetrated by 0.5% Triton X-100. Then they were incubated with prehybridization liquid, followed by incubation with the probes for lncCIRBIL. Immunofluorescence was analyzed under a confocal fluorescence microscope (Carl Zeiss, Germany).

**RNA-binding protein immunoprecipitation**. RIP was performed using a Magna RIP RNA-Binding Protein Immunoprecipitation Kit (Millipore, MA, USA) according to the manufacturer's protocol (Supplementary Data 2).

**RNA pulldown assay**. LncCIRBIL was subcloned into GV209 plasmid and purified using the QIA quick Gel Extraction Kit (QIAGEN, Lot: 163038791, Duesseldorf, Germany). Then the DNA was transcribed to RNA in vitro using the Roche Kit (Roche Diagnostics GmbH, Lot: 34035827, Mannheim, Germany). The obtained RNA was labeled with biotin (Roche Diagnostics GmbH, Lot: 31385421, Mannheim, Germany). Then the biotinylated RNAs were treated with RNase-free DNase I (Invitrogen, Lot: 2168762, Carlsbad, America) and purified with G50 Sephadex Quick Spin columns (Roche Diagnostics GmbH, Lot: 94016622, Mannheim, Germany). The cardiac tissue was lysed using lysis buffer supplemented with protease and phosphatase inhibitor cocktails and anti-RNase (Millipore, MA, USA). The streptavidin magnetic beads (Millipore, Lot: 2630691, MA, USA) were mixed with RNAs and proteins, and the mixture was shaken violently at a shaking table for 1 h. Then the mixture was centrifuged, and supernatant was discarded. The precipitate was eluted to obtain the RNA-binding-protein complex. The lncRNA-interacting proteins were further separated by SDS-PAGE and the gel was silver stained. Western blot was used to analyze the bound proteins. The antisense group was employed as a control.

**Mass spectrometry**. Mass spectrometry were performed at Beijing Protein Innovation Co. Ltd. In detail, bands of interest were cut off and washed with ddH$_2$O$_2$ for 10 min. The gel pieces were destained by vortexing in 1 mL of 25 mM NH$_4$HCO$_3$(ABC) in 50% acetonitrile for 20 min, then were dehydrated by acetonitrile and completely dried by Speedvac. Reduction was performed by adding 10 mM dithiothreitol (DTT) to gel pieces and incubating at 56 °C for 1 h. After removing all leftover DTT, alkylation was performed by incubating the gel pieces in 55 mM iodoacetamide (IAM) at room temperature in a dark room for 45 min. All leftover

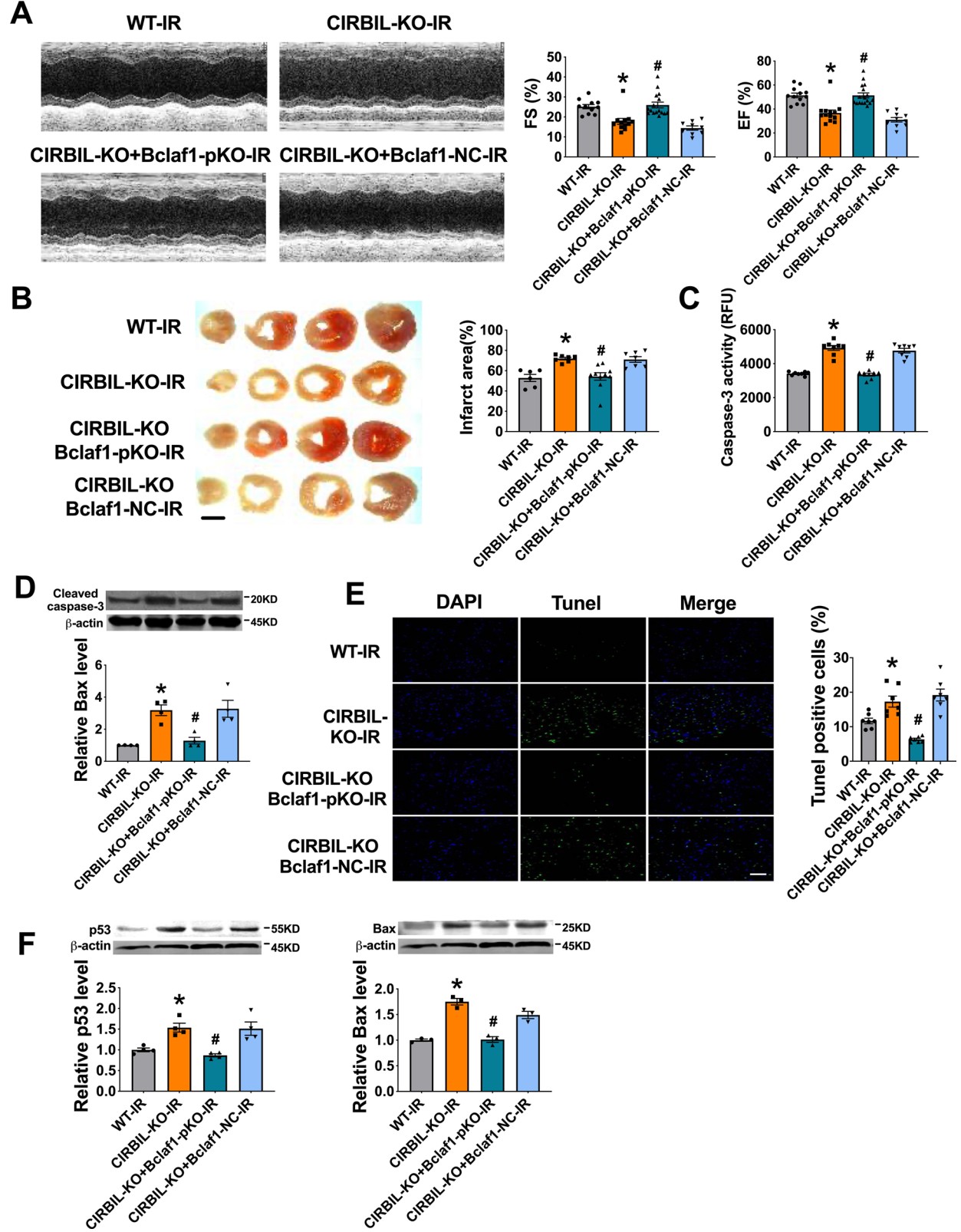

IAM was removed. Destaining, dehydration, and Speedvac steps were then repeated one time. Trypsin stock solution (1 µg/µL) was diluted 15 times by 25 mM ABC. The diluted trypsin solution was used to digest the gel pieces at 37 °C overnight. Formic acid (0.1%) was used to stop digestion on the next day. Non-extracted digests containing peptides (10 µL) were stored at −80 °C for further MS analysis.

For analysis of peptides, Thermo ScientificTM Q ExactiveTM high-resolution MS was coupled to UltiMate3000 RSLC nano ultra-high performance liquid system. Mobile phase A is 0.1% formic acid in water, mobile phase B is 0.1%

formic acid in 98% acetonitrile. Liquid chromatography (LC) flow rate was set at 0.4 mL/min with a gradient program as follows: 0–6 min = 5–8% B; 7–40 min = 8–30% B; 41–45 min = 30–60% B; 46–48 min = 60–80% B; 49–56 min = 80% B; 56–58 min = 80–5% B; 58–65 min = 5% B. Full scan spectra were acquired with a scan range of $m/z$ = 350–2000 and a resolution of 60,000. Data-dependent acquisition procedure were used for data acquisition. The 20 most intense precursor ions were then selected for HCD fragmentation. MS/MS spectra were acquired at a resolution of 17,500 with a normalized collision energy of 28%.

**Fig. 7 Partial knockout (pKO) of Bclaf1 abrogates the exacerbating effects of lncCIRBIL knockout on cardiac ischemia–reperfusion injury. A** Effects of Bclaf1 partial knockout on cardiac function of lncCIRBIL knockout mice subjected to ischemia/reperfusion injury. WT-I/R $N = 12$, CIRBIL-KO-I/R $N = 13$, CIRBIL-KO-Bclaf1-pKO-I/R $N = 17$, CIRBIL-KO-Bclaf1-NC-I/R $N = 10$. *$P < 0.05$ versus WT-IR, #$P < 0.05$ versus lncCIRBIL-KO-IR. $P$ values were determined by one-way ANOVA test followed by Bonferroni post hoc analysis. **B** Infarct size measured by TTC staining (scale bar: 2 mm). WT-I/R $N = 6$, CIRBIL-KO-I/R $N = 7$, CIRBIL-KO-Bclaf1-pKO-I/R $N = 10$, CIRBIL-KO-Bclaf1-NC-I/R $N = 7$. *$P < 0.05$ versus WT-IR, #$P < 0.05$ versus lncCIRBIL-KO-IR. $P$ values were determined by one-way ANOVA test followed by Bonferroni post hoc analysis. **C, D** Caspase-3 activity and cleaved caspase-3 protein level. $N = 8$ mice per group for caspase 3 activity, and $N = 3$ for cleaved caspase-3 protein level. *$P < 0.05$ versus WT-IR, #$P < 0.05$ versus lncCIRBIL-KO-IR. $P$ values were determined by one-way ANOVA test followed by Bonferroni post hoc analysis. **E** Cardiomyocyte apoptosis by TUNEL assay (scale bar: 20 μm). $N = 7$ mice per group. *$P < 0.05$ versus WT-IR, #$P < 0.05$ versus lncCIRBIL-KO-IR. $P$ values were determined by one-way ANOVA test followed by Bonferroni post hoc analysis. **F** The protein levels of p53 and Bax by western blot. $N = 3$ mice per group. *$P < 0.05$ versus WT-IR, #$P < 0.05$ versus lncCIRBIL-KO-IR. $P$ values were determined by one-way ANOVA test followed by Bonferroni post hoc analysis. Data are expressed as mean ± SEM.

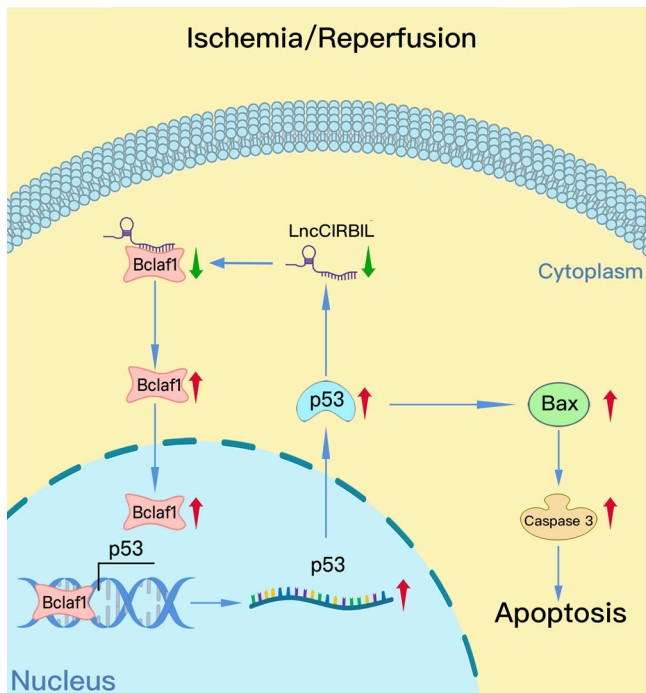

**Fig. 8 Schematic showing the proposed mechanism for the effects of lncCIRBIL on cardiac ischemia–reperfusion injury.** LncCIRBIL localizes in the cytoplasm of cardiomyocytes where it binds Bclaf1 protein to prevent its translocation to the nucleus. During cardiac ischemia–reperfusion injury, lncCIRBIL expression is reduced, which favors the dissociation of Bclaf1 protein from its binding state to become free for nuclear translocation. After entering the nucleus, Bclaf1 activates the transcription of p53 and Bax to induce cardiomyocyte apoptosis leading to cardiac ischemia–reperfusion injury. Meanwhile, p53 negatively regulates the transcription of lncCIRBIL and upregulation of p53 causes further downregulation of lncCIRBIL. Based on these findings, a novel positive feedback regulatory circuit is proposed: I/R or H/R → lncCIRBIL↓ → Bclaf1↑ → p53↑ → lncCIRBIL↓ → I/R or H/R injury↑.

The LC-MS raw data were processed using MASCOT2.3.0. MS/MS spectra were searched against uniport (20,160,315 entries) with trypsin-specific cleavage allowing up to 2 missing cleavages. Tolerance of precursor ions was set to 15 ppm and that of fragment ions was 20 mmu. Carbamidomethyl on Cys was fixed modification, and Gln->pyro-Glu (N-term Q) and oxidation on Met were variable modifications. Peptide ion score was set at >23; false discovery rate <0.05.

**Apoptosis assay**. Apoptosis was determined by TUNEL using an In-Situ Cell Death Detection Kit (Roche, Basel, Switzerland). The nuclei were stained with DAPI (Solarbio, Beijing, China). Immunofluorescence was analyzed under a fluorescence microscope (Carl Zeiss, Germany).

**CKMB detection**. The CKMB was detected by using the mouse CKMB (Creatine Kinase MB Isownzyme) Elisa Kit (Elabscience, Wuhan, China).

**LDH detection**. LDH level was detected by using the LDH Detection Kit (Nanjing Jiancheng Bioengineering Institute, Nanjing, China).

**Caspase-3 activity assay**. Caspase 3 activity was measured with a kit from Cell Signaling (catalog number #5723) according to the manufacturer's instruction.

**Statistical analysis**. Data are expressed as the mean ± SEM of at least three independent experiments for each experimental group. Student's *t* test was used for comparisons between two groups and one-way analysis of variance followed by Bonferroni corrected post hoc *t* test was used for multi-group comparisons (Graphpad Prism 8.0). A value of $P < 0.05$ was considered statistically significant.

**Reporting summary**. Further information on research design is available in the Nature Research Reporting Summary linked to this article.

## Data availability

The data that support the findings of this study and unique materials are available from the corresponding authors upon reasonable request. The lncCIRBIL microarray data are deposited to the Gene Expression Omnibus (GSE) with the accession number GSE161151. Additional data related to this paper may be requested from the authors. Source data are provided with this paper.

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

## Acknowledgements
This work was supported by National Key R&D Program of China (2017YFC1307404 to Z.P.), National Natural Science Foundation of China (81730012 and 81861128022 to B.Y. and 81903614 to Y. Zhang), Funds for Distinguished Young Scholars of Heilongjiang Province (to Z.P.), Heilongjiang Touyan Innovation Team Program (to B.Y.), and Yu Weihan Excellent Youth Foundation of Harbin Medical University (001000004 to Z.P.).

## Author contributions
Y. Zhang, X.Z., B.C., Y.J., Ying Li, and X.F. performed experiments, analyzed data, and prepared the manuscript. Y. Zhao, H.G., Y.Y., Jiming Yang, S.L., H.W., X.J., G.X., Jiqin Yang, W.M., Q.H., Yue Li, and T.T. helped perform experiments and collect data. B.Y. and Yanjie Lu oversaw the project and proofread the manuscript. Z.P. designed the project, oversaw the experiments, and prepared the manuscript.

## Competing interests
The authors declare no competing interests.
