## [Peer Review File · Nature Communications]

REVIEWER COMMENTS

Reviewer #1 (Remarks to the Author):

Short Name: LncRNA-CIRBIL regulates cardiac I/R injury via Bclaf1

The manuscript is overall very well written and explores the interaction between BCL2-associated transcription factor 1(Bclaf1) and lncRNA-CIRBIL and their combined effect on cardiac ischemia/reperfusion injury. The authors were able to determine that Bclaf1 was a downstream molecule of the lncRNA which, through sequestering the protein to the cytoplasm, allowed for cardio-protective action on the heart. This paper was able to expertly demonstrate both the overexpression and knockout effects of both molecules in vitro and in vivo to further validate the regulation of cardiac ischemia/reperfusion injury.

Major Points:

- This manuscript was not able to clearly identify which molecule acted on the other. Largely, the authors seemed to conclude that lncRNA-CIRBIL is the upstream molecule that acts on Bclaf1. However, there were instances where the wording appeared contradictory and Bclaf1 appeared to have an effect on lncRNA-CIRBIL. For example, the authors state "The interaction between lncRNA-CIRBIL and Bclaf1 indicates that Bclaf1 may be important in regulating the effects of lncRNA-CIRBIL on cardiac injury." This implies that the protein regulates the lncRNA. Another seemingly contradictory sentence seemed to be "These data further suggest that Bclaf1 is a downstream molecule that mediates the effects of lncRNA-CIRBIL on cardiac I/R injury." This implies that even though Bclaf1 is the downstream molecule, it can then control the effect lncRNA-CIRBIL has on cardiac injury. While this manuscript is extremely informative and the research is thoroughly extensive, clearly defining the relationship between both molecules is crucial; even if they do in fact act on each other (then this needs to be explicitly concluded). Figure 8 should seemingly help determine this relationship; however, it was also not so clear to help define the interaction. I would recommend that both the interaction and Figure 8 are changed to clearly understand how both molecules are acting upon each other.
- For the TUNEL data, the quantification was not so representative of the images. For example, Figure 6B shows a significant signal change between the OE and the NC samples while the quantification of this data is not representative of the images shown. Please provide images that more accurately depict quantification data.
- Can you please explain the reason for choosing this particular hypoxia/re-oxygenation model? No reference protocol is referred to in the material and methods section.
- In Figure 4, when Bclaf1 is overexpressed, an increase in apoptosis is shown, but was the nuclear expression recorded? It can be assumed that increased apoptosis levels are caused by a nuclear translocation of Bclaf1, but was this shown?
 - o Out of curiosity, was there a cardiac phenotype seen long-term in the mice after Bclaf1 overexpression?
- Is cleaved-Caspase3 WB data available instead of just a qPCR of Caspase3?
- Why is using the AAV9 virus considered a "knockdown" model when it uses CRISPR/Cas9? Why is not considered a partial "knockout" model? I understand that a small proportion of the cells may have the knockout, but is it still a knockout not a knockdown correct?

Minor Points:

- Please provide the rationale of measuring LDH and CKMB levels. It was not clearly stated anywhere why these levels were being recorded and it would be helpful to the reader to understand properly why these measurements were being taken.
- Figure 1: In the results section of the manuscript, a region is referred to as the remote area. In the figure itself as well as the figure legend, it is referred to as the NIZ. Please correct this inconsistency or mention somewhere in the manuscript that they are interchangeable.
- Figure 5A: Please provide uncropped WB image, the quality seen in the manuscript appears insufficient.

- Figure 6: What are the abbreviations OE and NC representing? None of these abbreviations were ever expressly stated and it makes understanding the data much more difficult. Please clearly explain exactly what each condition name presents in the figure legend or in the results explaining the figure.
 - o Some inconsistencies with labeling in Figure 6. In the TUNEL images a sample is named NC-HR and in all other locations it is labeled CIRBIL-NC-HR. Please change this and be consistent between TUNEL images and TUNEL positive cell quantifications.
- In general the English is very well written, but I would recommend a native speaker reading through it once briefly to edit minor points
 - o Ex. A sentence in the discussion seems like an incomplete thought as it reads: "The RNA pulldown and RIP assay reciprocally proved that lncRNA-CIRBIL directly interacts with."
 - o Further down in the same discussion paragraph you have written " One only example"
 - o Ex. CRISPR is spelled with an R missing as "CRISP" in the material and methods section
 - o There may be a spelling mistake in the graphs, Tublin is written where I assume Tubulin was meant? And this error was consistent throughout the WB data.
 - o In vivo and in vitro should always be italicized
 - o All abbreviations should be introduced at least once

Reviewer #2 (Remarks to the Author):

The manuscript entitled "Disruption BCL2-associated transcription factor (Bclaf1) nuclear translation by lncRNA-CIRBIL alleviates cardiac ischemia/reperfusion injury" by Zhang et al. characterizes a novel pro-apoptotic pathway involved in I/R injury, lncRNA-CIRBIL-Bclaf1-p53. The study was very thorough involving gain-of-function and loss-of-function experiments of lncRNA-CIRBIL or Bclaf1 in both cell and mouse models of I/R. Zhang et al. identified that lncRNA-CIRBIL becomes downregulated after I/R in mice, which is then unable to sequester pro-apoptotic transcription factor Bclaf1 to the cytoplasm. Bclaf1 translocation to the nucleus was able to induce p53 and Bax transcription to promote apoptosis. This pathway was well illustrated in Fig. 8. I have a few suggestions for improvement:

Major comments:

1. Given that Bclaf1 is not well known in its role in cardiac diseases, but has been established in cancer and other processes, consider expanding the discussion on what is known about Bclaf1 especially in terms of apoptosis and how it may compare to your studies. Do the pathways generally remain the same? Is there any information on how lncRNA-CIRBIL is regulated or how it is reduced during I/R or any other disease state?
 - a. Further, from IHC images in Fig. 3F, it looks like Bclaf1 in general is increased in expression throughout the cell in WT-IR compared to WT suggesting Bclaf1 expression is induced with I/R injury also. This could also contribute to increased injury, even if CIRBIL levels were to remain normalized?
2. The mouse model of I/R involves true ischemia, while the cell models only involve hypoxia. The supplemental data do confirm/correlate well with Zhang et al's in vivo findings, but I was curious if you had confirmed with nutrient starvation along with hypoxia. Can we speculate that the stimuli for reduced lncRNA-CIRBIL is hypoxia vs ischemia? Is there anything in the cancer field connecting hypoxia and Bclaf1?
3. If it is possible, it may be useful to see some sort of correlative analysis between degree of infarct size and extent of downregulation of lncRNA-CIRBIL (or cardiac function...which ever readouts are available from the same mice). Could lncRNA-CIRBIL can be used as some sort of prognosis marker?

Minor comments:

1. Do you have any Western blot analysis of cytosolic fractions to go along with the data where you show nuclear enrichment/inhibition of Bclaf1? (Fig. 3D-G) If available consider adding these, they would complement the nuclear fractions.

2. Consider correcting the following in the manuscript:

a. Line 50: "diverse class of RNAs that is more" ◊ needs plural verb

b. Line 92: "deregulated lncRNAs that is downregulated" ◊ needs plural verb

c. Line 246: "ischemia reperfusion(I/R) injury" ◊ missing a space

d. Line 304: "...that interacts with Bcl2, which mainly located" ◊ missing verb ie "was"

e. Line 318: "...knockout of lncRNA-CIRBIL drove them to the nucleus" ◊ change "them" to "it"

f. Line 320: "underlies the pathogenesis of cardiac I/R injury" ◊ perhaps use "contributes" vs "underlies", as downregulation of lncRNA-CIRBIL is not the only pro-apoptotic mechanism during I/R injury.

g. Line 341: "After 24 hours' reperfusion" ◊ delete apostrophe

h. Line 413: "determine eject fraction" ◊ change "eject" to "ejection"

i. Line 498: "and the mixture were shaking.." ◊ change "were" to "was"

3. Certain merged DAPI and TUNEL immunofluorescence images make it difficult to see where TUNEL (green) is overlapping with DAPI. For example in Fig. 2G in the merged image for CIRBIL-KO-IR, the green fluorescence looks like it's disappeared compared to ample green TUNEL+ cells in TUNEL alone in CIRBIL-KO-IR. Same with the merged image for CIRBIL-SI Bclaf1-NC-HR in Fig. 6F.

4. Consider selecting a more even representative WB image for Lamin loading control in Fig. 3E.

5. I was curious if you confirmed CIRBIL is still downregulated or not in your Bclaf1-KO-IR mice?

6. If it is possible, is there a way to confirm CIRBIL is also downregulated in the human heart during heart disease ie heart failure? Perhaps there is deposited lncRNA seq data on human tissues you could assess? I think this would nicely complement your animal/cell studies.

7. Please clarify in the figure legend what area of the heart TUNEL staining sections were quantified from (BZ?). (Fig. 1I, 2G, 4G, 5G)

REVIEWER COMMENTS

Reviewer #1 (Remarks to the Author):

Short Name: LncRNA-CIRBIL regulates cardiac I/R injury via Bclaf1

The manuscript is overall very well written and explores the interaction between BCL2-associated transcription factor 1(Bclaf1) and lncRNA-CIRBIL and their combined effect on cardiac ischemia/reperfusion injury. The authors' were able to determine that Bclaf1 was a downstream molecule of the lncRNA which, through sequestering the protein to the cytoplasm, allowed for cardio-protective action on the heart. This paper was able to expertly demonstrate both the overexpression and knockout effects of both molecules in vitro and in vivo to further validate the regulation of cardiac ischemia/reperfusion injury.

Reply: We would like to express our sincere gratefulness to you for your positive comments on our work, and your suggestions are constructive and very helpful for improving the quality of our manuscript. Please find our point-by-point responses to your individual comments and suggestions as shown below.

Major Points:

- This manuscript was not able to clearly identify which molecule acted on the other. Largely, the authors seemed to conclude that lncRNA-CIRBIL is the upstream molecule that acts on Bclaf1. However, there were instances where the wording appeared contradictory and Bclaf1 appeared to have an effect on lncRNA-CIRBIL. For example, the authors' state "The interaction between lncRNA-CIRBIL and Bclaf1 indicates that Bclaf1 may be important in regulating the effects of lncRNA-CIRBIL on cardiac injury." This implies that the protein regulates the lncRNA. Another seemingly contradictory sentence seemed to be "These data further suggest that Bclaf1 is a downstream molecule that mediates the effects of lncRNA-CIRBIL on cardiac I/R injury." This implies that even though Bclaf1 is the downstream molecule, it can then control the effect lncRNA-CIRBIL has on cardiac injury. While this

manuscript is extremely informative and the research is thoroughly extensive, clearly defining the relationship between both molecules is crucial; even if they do in fact act on each other (then this needs to be explicitly concluded). Figure 8 should seemingly help determine this relationship; however, it was also not so clear to help define the interaction. I would recommend that both the interaction and Figure 8 are changed to clearly understand how both molecules are acting upon each other.

Reply: Thank you so much for the insightful comment. We realized that it was not very clear for the description and the schematic graph for interaction of lnc-CIRBIL and bclaf1. In fact, our data showed that lnc-CIRBIL can directly bind to bclaf1 and is reduced in the cytosol of cardiac myocytes during cardiac ischemia/reperfusion injury. Thereby, the reduced lnc-CIRBIL will result in less binding to bclaf1 and the lnc-CIRBIL-free bclaf1 in the cytosol will be increased, which facilitates the nuclear translocation of bclaf1 and leads to its increase in the nucleus. We have revised both the description and the schematic graph in the revised manuscript.

• For the TUNEL data, the quantification was not so representative of the images. For example, Figure 6B shows a significant signal change between the OE and the NC samples while the quantification of this data is not representative of the images shown. Please provide images that more accurately depict quantification data.

Reply: Thank you for the good suggestion. We have replaced the images with better ones as follows.

• Can you please explain the reason for choosing this particular hypoxia/re-oxygenation model? No reference protocol is referred to in the material and methods section.

Reply: Thank you for your comment. In the present study, we used ischemia/reperfusion model to investigate lncRNA-CIRBIL role in cardiac injury in vivo. It is widely accepted that cellular hypoxia/reoxygenation model to mimic the ischemia/reperfusion processes in vivo¹⁻³. We improperly described it in the “**Cell transfection with lncRNA-CIRBIL plasmids or siRNA**” method subsection. As suggested, we have included this information in the section “**Mouse neonatal cardiomyocyte isolation and culture**”.

References:

1. Gu S, Tan J, Li Q, Liu S, Ma J, Zheng Y, Liu J, Bi W, Sha P, Li X, Wei M, Cao N, Yang HT. Downregulation of LAPTM4B Contributes to the Impairment of the Autophagic Flux via Unopposed Activation of mTORC1 Signaling During Myocardial Ischemia/Reperfusion Injury. *Circ Res.* 2020 Sep 11;127(7):e148-e165.
2. Fazal L, Laudette M, Paula-Gomes S, Pons S, Conte C, Tortosa F, Sicard P, Sainte-Marie Y, Bissierier M, Lairez O, Lucas A, Roy J, Ghaleh B, Fauconnier J, Mialet-Perez J, Lezoualc'h F. Multifunctional Mitochondrial Epac1 Controls Myocardial Cell Death. *Circ Res.* 2017 Feb 17;120(4):645-657.
3. Turner MS, Haywood GA, Andreka P, You L, Martin PE, Evans WH, Webster KA, Bishopric NH. Reversible connexin 43 dephosphorylation during hypoxia and reoxygenation is linked to cellular ATP levels. *Circ Res.* 2004 Oct 1;95(7):726-33.

• In Figure 4, when Bclaf1 is overexpressed, an increase in apoptosis is shown, but was the nuclear expression recorded? It can be assumed that increased apoptosis levels are caused by a nuclear translocation of Bclaf1, but was this shown?

Reply: Thank you for the good suggestion. Accordingly, we examined the nuclear expression of Bclaf1 by immunofluorescent staining in Bclaf1 transgenic mice. We found that the level of bclaf1 was increased in both the cytoplasm and nucleus of cardiomyocytes from Bclaf1 transgenic mice than that of wildtype controls. Bclaf1 was increased in the nucleus of wildtype cardiomyocytes during cardiac ischemia/reperfusion injury, which was more remarkable in the Bclaf1 transgenic mice. The data was included in Figure 4.

o Out of curiosity, was there a cardiac phenotype seen long-term in the mice after Bclaf1 overexpression?

Reply: Thank you for the good suggestion. Cardiac function was performed in 4-month old Bclaf1 tg mice. We found that the EF and FS were significantly reduced compared with wild-type littermates. It can be speculated Bclaf1 overexpression possesses other deteriorating effects likely via different mechanism in the heart, which deserves to further investigation.

• Is cleaved-Caspase3 WB data available instead of just a qPCR of Caspase3?

Reply: Thank you for the good suggestion. Accordingly, we have collected all the WB data of cleaved-caspase 3 and included them in the related Figures (Fig1,2, 4-7, Supple Fig 4, 6,8,9).

- Why is using the AAV9 virus considered a “knockdown” model when it uses CRISPR/Cas9? Why is not considered a partial “knockout” model? I understand that a small proportion of the cells may have the knockout, but is it still a knockout not a knockdown correct?

Reply: Thank you for the good suggestion. We totally agree with your opinion. The information on this technology AAV9-carrying CRISPR/Cas9 is not correct. We have corrected the description as “partial knockout” in the main text.

Minor Points:

- Please provide the rational of measuring LDH and CKMB levels. It was not clearly stated anywhere why these levels were being recorded and it would be helpful to the reader to understand properly why these measurements were being taken.

Reply: Thank you for the good suggestion. LDH and CKMB are enzymes residing in the cytoplasm of cardiomyocytes, and are released into the plasma upon cardiomyocyte injury. We have added this information in the Result part when LDH and CKMB data is mentioned.

- Figure 1: In the results section of the manuscript, a region is referred to as the remote area. In the figure itself as well as the figure legend, it is referred to as the NIZ. Please correct this inconsistency or mention somewhere in the manuscript that they are interchangeable.

Reply: Thank you for the good suggestion. We have changed the remote area to NIZ(remote non-ischemia zone) in the results section.

- Figure 5A: Please provide uncropped WB image, the quality seen in the manuscript appears insufficient.

Reply: Thank you for the suggestion. Please see the uncropped WB image below. To avoid any confusion, we repeated this experiment and replaced the WB bands with

high quality ones. The sample number was increased to six. The uncropped WB images and the revised Figure were shown below.

A. The original figure and uncropped WB images.

B. The revised figure.

• Figure 6: What are the abbreviations OE and NC representing? None of these abbreviations were ever expressly stated and it makes understanding the data much more difficult. Please clearly explain exactly what each condition name presents in the figure legend or in the results explaining the figure.

Reply: Thank you for the comment. OE represents for LncRNA-CIRBIL overexpressing plasmid, and NC for Negative control (Empty plasmid). We have added this information in the figure legend of Figure 6.

o Some inconsistencies with labeling in Figure 6. In the TUNEL images a sample is named NC-HR and in all other locations it is labeled CIRBIL-NC-HR. Please change this and be consistent between TUNEL images and TUNEL positive cell quantifications.

Reply: We feel so sorry for this mistake. We have corrected NC-HR to CIRBIL-NC-HR.

• In general the English is very well written, but I would recommend a native speaker reading through it once briefly to edit minor points

Reply: Thank you for the good suggestion. We have invited an English speaker to read through the manuscript and perform certain revision.

o Ex. A sentence in the discussion seems like an incomplete thought as it reads: “The RNA pulldown and RIP assay reciprocally proved that lncRNA-CIRBIL directly interacts with.”

Reply: We apologize for this mistake. This is an incomplete sentence. It should be “The RNA pulldown and RIP assay reciprocally proved that lncRNA-CIRBIL directly binds to Bclaf1”.

o Further down in the same discussion paragraph you have written “ One only example”

Reply: We apologize for this mistake. It should be “The only example”.

o Ex. CRISPR is spelled with an R missing as “CRISP” in the material and methods section

Reply: Thank you for the reminding. As suggested, we have corrected it.

o There may be a spelling mistake in the graphs, Tublin is written where I assume Tubulin was meant? And this error was consistent throughout the WB data.

Reply: Sorry for the mistake. It is “tubulin”. We have corrected it.

o In vivo and in vitro should always be italicized

Reply: Sorry for the mistake. We have corrected them all.

o All abbreviations should be introduced at least once

Reply: We have checked the whole manuscript and corrected them all.

Reviewer #2 (Remarks to the Author):

The manuscript entitled “Disruption BCL2-associated transcription factor (Bclaf1) nuclear translation by lncRNA-CIRBIL alleviates cardiac ischemia/reperfusion injury” by Zhang et al. characterizes a novel pro-apoptotic pathway involved in I/R injury, lncRNA-CIRBIL-Bclaf1-p53. The study was very thorough involving gain-of-function and loss-of-function experiments of lncRNA-CIRBIL or Bclaf1 in both cell and mouse models of I/R. Zhang et al. identified that lncRNA-CIRBIL becomes downregulated after I/R in mice, which is then unable to sequester pro-apoptotic transcription factor Bclaf1 to the cytoplasm. Bclaf1 translocation to the nucleus was able to induce p53 and Bax transcription to promote apoptosis. This pathway was well illustrated in Fig. 8. I have a few suggestions for improvement:

Reply: We are grateful to you for your very positive comments on our work and constructive suggestions for improving our manuscript. We have performed several sets of additional experiments to address your concerns.

Major comments:

1. Given that Bclaf1 is not well known in its role in cardiac diseases, but has been established in cancer and other processes, consider expanding the discussion on what is known about Bclaf1 especially in terms of apoptosis and how it may compare to your studies. Do the pathways generally remain the same?

Reply: Thank you for the good suggestion. We have revised this issue as suggested (Page 17, Line1-13).

Bclaf1 was originally validated as an apoptosis inducer by activating p53 pathway. We observed the same phenomenon in cardiac ischemia/reperfusion injury. However, recent studies also demonstrated the oncogenic role of Bclaf1 in hepatocellular carcinoma by regulating MYC Proto-Oncogene c-MYC mRNA

Stability and HeLa cells by enhancing hypoxia-inducible factor-1 α stability

(references). The discrepancy may be explained by the diverse biological property of various types of cancer cells or/and multiple mechanisms involved in lncRNA function.

Is there any information on how lncRNA-CIRBIL is regulated or how it is reduced during I/R or any other disease state?

Reply: Thank you for the insightful comment. This is an important issue that needs to be addressed. As lncRNA-CIRBIL is an intergenic lncRNA, we therefore analyzed the potential transcriptional factors that may regulate its expression. We found that p53 is a candidate transcriptional factor of lncRNA-CIRBIL. We then validated the regulation of p53 on lncRNA-CIRBIL expression. We found that overexpression of p53 in cultured cardiomyocytes reduced the expression of lncRNA-CIRBIL (Figure A). Knockdown of p53 in normal cultured cardiomyocytes did not change the expression of lncRNA-CIRBIL (Figure B), while it upregulated the expression of lncRNA-CIRBIL when the cells were exposed to hypoxia/reoxygenation (Figure C). The reason that p53 siRNA did not change the expression of lncRNA-CIRBIL may be that the basal p53 level is very low, which is hardly affected by its siRNA in the nucleus and is mainly distributed in the cytoplasm in cardiomyocytes. These data indicated that lncRNA-CIRBIL is negatively regulated by p53. LncRNA-CIRBIL, BCLAF1 and p53 form a positive feedback circuit. These data were presented in supplementary Figure 9.

a. Further, from IHC images in Fig. 3F, it looks like Bclaf1 in general is increased in expression throughout the cell in WT-IR compared to WT suggesting Bclaf1 expression is induced with I/R injury also. This could also contribute to increased injury, even if CIRBIL levels were to remain normalized?

Reply: Thank you for the insightful comment. In this study we found that Bclaf1 was induced by I/R injury, and it contributes to cardiac injury as shown by the data obtained from bclaf1-tg mice subjected to I/R injury. We agree that increased expression of Bclaf1 also contributes to the elevated nuclear level of Bclaf1. CIRBIL can sequester the increased Bclaf1 in the cytoplasm and reduce the nuclear level of Bclaf1 during cardiac I/R injury, which is considered the main mechanism for the influence of CIRBIL on cardiac injury.

2. The mouse model of I/R involves true ischemia, while the cell models only involve hypoxia. The supplemental data do confirm/correlate well with Zhang et al's in vivo findings, but I was curious if you had confirmed with nutrient starvation along with hypoxia. Can we speculate that the stimuli for reduced lncRNA-CIRBIL is hypoxia vs ischemia?

Reply: Thank you for the good suggestion. As suggested, we examined the expression of lncRNA-CIRBIL in cardiomyocytes exposed hypoxia, hypoxia+ nutrient starvation(low glucose) and H₂O₂. The data showed that lncRNA-CIRBIL was reduced under all the three different conditions and was more pronounced in

hypoxia+ nutrient starvation(low glucose, LG) group. These data were presented in supplementary Figure 3B-D. NG, normal glucose. Should let reviewer believe hypoxia/xx model used in the study is well accepted model to mimic I/R injury. This question was asked by reviewer 1

Is there anything in the cancer field connecting hypoxia and Bclaf1?

Reply: A recently published paper showed that bclaf1 is a direct transcriptional target of HIF-1 and upregulated in multiple cell lines during hypoxia (Oncogene. 2020 Mar;39(13):2807-2818), which is consistent with our finding in the study. They reported that bclaf1 promoted tumor cell growth, while in this study increased bclaf1 injured cardiac myocytes. We think that the functional discrepancy of bclaf1 may be caused by the diverse biological property in different cell types.

3. If it is possible, it may be useful to see some sort of correlative analysis between degree of infarct size and extent of downregulation of lncRNA-CIRBIL (or cardiac function...which ever readouts are available from the same mice). Could lncRNA-CIRBIL can be used as some sort of prognosis marker?

Reply: Thank you for the valuable comment. It is difficult to examine infarct size and the cardiac expression of lncRNA-CIRBIL in the same mouse heart. We then analyzed the correlation between cardiac lncRNA-CIRBIL level and cardiac function of ischemia/reperfusion mice. We discovered a negative correlation between lncRNA-CIRBIL level and cardiac function, indicating that the lower level of

lncRNA-CIRBIL, the severer cardiac injury. We completely agree that it will be very interesting to identify the prognostic value of lncRNA-CIRBIL in cardiac ischemia patients. We hope to explore this promising potential of lncRNA-CIRBIL in clinical implication in the future work.

Minor comments:

1. Do you have any Western blot analysis of cytosolic fractions to go along with the data where you show nuclear enrichment/inhibition of Bclaf1? (Fig. 3D-G) If available consider adding these, they would complement the nuclear fractions.

Reply: Thank you for the good suggestion. We have added these data as suggested.

2. Consider correcting the following in the manuscript:

- a. Line 50: “diverse class of RNAs that is more” ◇ needs plural verb
- b. Line 92: “deregulated lncRNAs that is downregulated” ◇ needs plural verb
- c. Line 246: “ischemia reperfusion(I/R) injury” ◇ missing a space
- d. Line 304: “..that interacts with Bcl2, which mainly located” ◇ missing verb ie “was”
- e. Line 318: “...knockout of lncRNA-CIRBIL drove them to the nucleus” ◇ change “them” to “it”
- f. Line 320: “underlies the pathogenesis of cardiac I/R injury” ◇ perhaps use “contributes” vs “underlies”, as downregulation of lncRNA-CIRBIL is not the only pro-apoptotic mechanism during I/R injury.
- g. Line 341: “After 24 hours’ reperfusion” ◇ delete apostrophe
- h. Line 413: “determine eject fraction” ◇ change “eject” to “ejection”
- i. Line 498: “and the mixture were shaking..” ◇ change “were” to “was”

Reply: We apologize for all these errors. We have corrected them all. Thanks for your reminding.

3. Certain merged DAPI and TUNEL immunofluorescence images make it difficult to see where TUNEL (green) is overlapping with DAPI. For example in Fig. 2G in the

merged image for CIRBIL-KO-IR, the green fluorescence looks like it's disappeared compared to ample green TUNEL+ cells in TUNEL alone in CIRBIL-KO-IR. Same with the merged image for CIRBIL-SI Bclaf1-NC-HR in Fig. 6F.

Reply: We apologize for the poor-quality images. As suggested, we have performed these staining assays and obtained better images. We have replaced the images with better ones.

4. Consider selecting a more even representative WB image for Lamin loading control in Fig. 3E.

Reply: Thank you for the good suggestion. We have replaced WB image for Lamin loading control in Fig. 3E with a better one.

5. I was curious if you confirmed CIRBIL is still downregulated or not in your Bclaf1-KO-IR mice?

Reply: Thank you for the good suggestion. As suggested, we examined the level of CIRBIL in Bclaf1-KO-IR mice. Interestingly, the reduction of CIRBIL in WT-IR mice is partially restored by Bclaf1 knockout. This data was presented in supplementary Figure 9.

For the above finding, this may be explained that p53 can inhibit the expression of CIRBIL. We found that p53 is a negative regulator of lncRNA-CIRBIL. Overexpression of p53 in cultured cardiomyocytes reduced the expression of lncRNA-CIRBIL (Figure A). Knockdown of p53 in normal cultured cardiomyocytes did not change the expression of lncRNA-CIRBIL (Figure B), while it upregulated the expression of lncRNA-CIRBIL when the cells were exposed to hypoxia/reoxygenation (Figure C). The reason that p53 siRNA did not change the expression of lncRNA-CIRBIL may be that the basal p53 level is very low and is mainly distributed in the cytoplasm in cardiomyocytes. These data indicated that lncRNA-CIRBIL is negatively regulated by p53. lncRNA-CIRBIL, BCLAF1 and p53 form a positive feedback circuit. Therefore, when Bclaf1 is reduced, p53 will be inhibited, and its suppression on CIRBIL expression will be lessened and the level of CIRBIL will increase.

These data were presented in supplementary Figure 9.

6. If it is possible, is there a way to confirm CIRBIL is also downregulated in the human heart during heart disease ie heart failure? Perhaps there is deposited lncRNA seq data on human tissues you could assess? I think this would nicely complement your animal/cell studies.

Reply: Thank you for the good suggestion. We failed to obtain human cardiac tissues. Instead, we examined the level of human lncRNA-CIRBIL in the plasma of cardiac infarction patients. By sequence alignment, we found a segment of conservative sequence (red line) of mouse lncRNA-CIRBIL in human genome.

We then detected this sequence (human lncRNA-CIRBIL) in human plasma sample. We found that the level of human lncRNA-CIRBIL was dramatically reduced in the plasma of acute myocardial infarction (AMI) patients than in non-AMI subjects.

7. Please clarify in the figure legend what area of the heart TUNEL staining sections were quantified from (BZ?). (Fig. 1I, 2G, 4G, 5G)

Reply: Thank you for the good suggestion. Actually, we performed the TUNEL staining on sections from the border zone (BZ). We have indicated the specific region in these figure legends.

REVIEWERS' COMMENTS

Reviewer #1 (Remarks to the Author):

Manuscript Number: NCOMMS-20-14579A

Short Name: LncRNA-CIRBIL regulates cardiac I/R injury via Bclaf1

Second Revision:

The manuscript seems to be thoroughly revised and each point was directly answered by the authors. I greatly appreciate the detail that most questions were addressed. More experiments were performed when necessary and this expansion of data helps support the main hypothesis. I only had minor edits to report for this revision.

While most figures and legends seem to be edited appropriately for additions of new experiments, Figure 7 F still reads 7E in the legend and the TUNEL positive picture reads 7D instead of 7E in the legend. Please make the necessary adjustments.

A small comment, I appreciate that the authors changed the majority of places where Bclaf1 knockdown now properly reads partial knockout due to the AAV9-Cas9 system, however the title for "Knockdown of Bclaf1 abrogated the exacerbating effects of..." remains unchanged and the reference to this experiment in the introduction also still reads knockdown. Please make these two small edits for consistency.

The English does seem to have been improved overall; however, an additional quick read-through especially for the newly added/edited sections could only improve the manuscript before final submission. A small example, please see the additions under the "Knockdown of Bclaf1 abrogated the exacerbating effects of LncRNA-CIRCIL knockout..." In some instances, it is correct to say "we partially knocked out the expression..." but the following change of "mitigated by partial deletion of Bclaf1" should just be "partial" without the "ly" and this mistake was made a few times with this word. The native speaker can just quickly check any revised section of the manuscript so that it reads very easily.

Reviewer #2 (Remarks to the Author):

nothing further -

REVIEWERS' COMMENTS

Reply: Thank you so much for all the insightful comments and suggestions. They are very important for us to improve the quality of the work. We also learned a lot from revising the manuscript.

Reviewer #1 (Remarks to the Author):

Manuscript Number: NCOMMS-20-14579A

Short Name: LncRNA-CIRBIL regulates cardiac I/R injury via Bclaf1

Second Revision:

The manuscript seems to be thoroughly revised and each point was directly answered by the authors. I greatly appreciate the detail that most questions were addressed. More experiments were performed when necessary and this expansion of data helps support the main hypothesis. I only had minor edits to report for this revision.

While most figures and legends seem to be edited appropriately for additions of new experiments, Figure 7 F still reads 7E in the legend and the TUNEL positive picture reads 7D instead of 7E in the legend. Please make the necessary adjustments.

Reply: We apologize for the mistake. We have corrected them.

A small comment, I appreciate that the authors changed the majority of places where Bclaf1 knockdown now properly reads partial knockout due to the AAV9-Cas9 system, however the title for “Knockdown of Bclaf1 abrogated the exacerbating effects of...” remains unchanged and the reference to this experiment in the introduction also still reads knockdown. Please make these two small edits for consistency.

Reply: We apologize for failing to correct them in R1. We have corrected them.

The English does seem to have been improved overall; however, an additional quick read-through especially for the newly added/edited sections could only improve the manuscript before final submission. A small example, please see the additions under the “Knockdown of Bclaf1 abrogated the exacerbating effects of lncRNA-CIRCIL knockout...” In some instances, it is correct to say “we partially knocked out the expression...” but the following change of “mitigated by partially deletion of Bclaf1” should just be “partial” without the “ly” and this mistake was made a few times with this word. The native speaker can just quickly check any revised section of the manuscript so that it reads very easily.

Reply: We apologize for the grammar mistakes in English. We want to thank you for your patience and kindness. We have corrected the ones you pointed out. In addition, we invited one more native English speaker to read through the manuscript and edit the English, which have be highlighted in tracking mode.

Reviewer #2 (Remarks to the Author):

nothing further –

Reply: Thank you very much.